# Bootstrapping the Expressivity with Model-based Planning

## Abstract

We compare the model-free reinforcement learning with the model-based approaches through the lens of the expressive power of neural networks for policies, $Q$-functions, and dynamics. We show, theoretically and empirically, that even for one-dimensional continuous state space, there are many MDPs whose optimal $Q$-functions and policies are much more complex than the dynamics. We hypothesize many real-world MDPs also have a similar property. For these MDPs, model-based planning is a favorable algorithm, because the resulting policies can approximate the optimal policy significantly better than a neural network parameterization can, and model-free or model-based policy optimization rely on policy parameterization. Motivated by the theory, we apply a simple multi-step model-based bootstrapping planner (BOOTS) to bootstrap a weak $Q$-function into a stronger policy. Empirical results show that applying BOOTS on top of model-based or model-free policy optimization algorithms at the test time improves the performance on MuJoCo benchmark tasks.

## 1 Introduction

Model-based deep reinforcement learning (RL) algorithms offer a lot of potentials in achieving significantly better sample efficiency than the model-free algorithms for continuous control tasks. We can largely categorize the model-based deep RL algorithms into two types: 1. model-based policy optimization algorithms which learn policies or $Q$-functions, parameterized by neural networks, on the estimated dynamics, using off-the-shelf model-free algorithms or their variants (Luo et al., 2019; Janner et al., 2019; Kaiser et al., 2019; Kurutach et al., 2018; Feinberg et al., 2018; Buckman et al., 2018), and 2. model-based planning algorithms, which plan with the estimated dynamics Nagabandi et al. (2018); Chua et al. (2018); Wang & Ba (2019).

A deeper theoretical understanding of the pros and cons of model-based and the model-free algorithms in the continuous state space case will provide guiding principles for designing and applying new sample-efficient methods. The prior work on the comparisons of model-based and model-free algorithms mostly focuses on their sample efficiency gap, in the case of tabular MDPs (Zanette & Brunskill, 2019; Jin et al., 2018), linear quadratic regulator (Tu & Recht, 2018), and contextual decision process with sparse reward (Sun et al., 2019).

In this paper, we theoretically compare model-based RL and model-free RL in the continuous state space through the lens of *approximability* by neural networks, and then use the insight to design practical algorithms. What is the representation power of neural networks for expressing the $Q$-function, the policy, and the dynamics? How do the model-based and model-free algorithms utilize the expressivity of neural networks?

Our main finding is that even for the case of one-dimensional continuous state space, there can be a massive gap between the approximability of $Q$-function and the policy and that of the dynamics:

> The optimal $Q$-function and policy can be significantly more complex than the dynamics.

We construct environments where the dynamics are simply piecewise linear functions with constant pieces, but the optimal $Q$-functions and the optimal policy require an exponential (in the horizon)

---
* indicates equal contribution

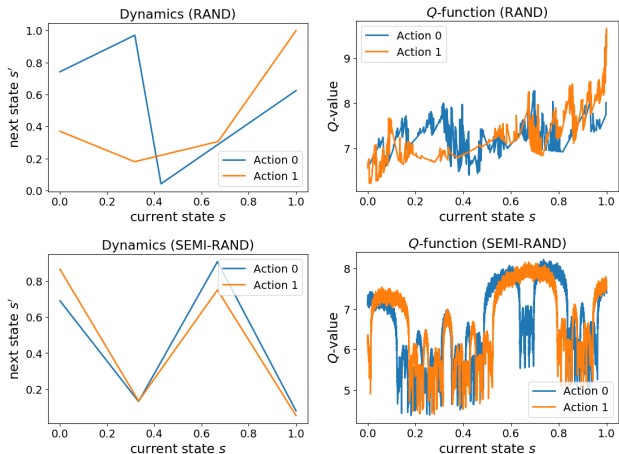

Figure 1: **Left:** The dynamics of two randomly generated MDPs (from the RAND, and SEMI-RAND methods outlined in Section 4.3 and detailed in Appendix D.1). **Right:** The corresponding $Q$-functions which are more complex than the dynamics (more details in Section 4.3).

number of linear pieces, or exponentially wide neural networks, to approximate.[1] The approximability gap can also be observed empirically on (semi-) randomly generated piecewise linear dynamics with a decent chance. (See Figure 1 for two examples.)

When the approximability gap occurs, any deep RL algorithms with policies parameterized by neural networks will suffer from a sub-optimal performance. These algorithms include both model-free algorithms such as DQN (Mnih et al., 2015) and SAC (Haarnoja et al., 2018), and model-based policy optimization algorithms such as SLBO (Luo et al., 2019) and MBPO (Janner et al., 2019). To validate the intuition, we empirically apply these algorithms to the constructed or the randomly generated MDPs. Indeed, they fail to converge to the optimal rewards even with sufficient samples, which suggests that they suffer from the lack of expressivity.

However, in such cases, model-based planning algorithms should not suffer from the lack of expressivity, because they only use the learned, parameterized dynamics, which are easy to express. The policy obtained from the planning is the maximizer of the total future reward on the learned dynamics, and can have an exponential (in the horizon) number of pieces even if the dynamics has only a constant number of pieces. In fact, even a partial planner can help improve the expressivity of the policy. If we plan for $k$ steps and then resort to some $Q$-function for estimating the total reward of the remaining steps, we can obtain a policy with $2^k$ more pieces than what $Q$-function has.

We hypothesize that the real-world continuous control tasks also have a more complex optimal $Q$-function and a policy than the dynamics. The theoretical analysis of the synthetic dynamics suggests that a model-based few-steps planner on top of a parameterized $Q$-function will outperform the original $Q$-function because of the addtional expressivity introduced by the planning. We empirically verify the intuition on MuJoCo benchmark tasks. We show that applying a model-based planner on top of $Q$-functions learned from model-based or model-free policy optimization algorithms in the test time leads to significant gains over the original $Q$-function or policy.

In summary, our contributions are:

1. We construct continuous state space MDPs whose $Q$-functions and policies are proved to be more complex than the dynamics (Sections 4.1 and 4.2.)

2. We empirically show that with a decent chance, (semi-)randomly generated piecewise linear MDPs also have complex $Q$-functions (Section 4.3.)

3. We show theoretically and empirically that the model-free RL or model-based policy optimization algorithms suffer from the lack of expressivity for the constructed MDPs (Sections 4.3), whereas model-based planning solve the problem efficiently (Section 5.2.)

4. Inspired by the theory, we propose a simple model-based bootstrapping planner (BOOTS), which can be applied on top of any model-free or model-based $Q$-learning algorithms at

---

[1]In turn, the dynamics can also be much more complex than the $Q$-function. Consider the following situation: a subset of the coordinates of the state space can be arbitrarily difficult to express by neural networks, but the reward function can only depend on the rest of the coordinates and remain simple.

the test time. Empirical results show that BOOTS improves the performance on MuJoCo benchmark tasks, and outperforms previous state-of-the-art on MuJoCo humanoid environment.

## 2 ADDITIONAL RELATED WORK

**Comparisons with Prior Theoretical Work.** Model-based RL has been extensively studied in the tabular case (see (Zanette & Brunskill, 2019; Azar et al., 2017) and the references therein), but much less so in the context of deep neural networks approximators and continuous state space. (Luo et al., 2019) give sample complexity and convergence guarantees suing principle of optimism in the face of uncertainty for non-linear dynamics.

Below we review several prior results regarding model-based versus model-free dichotomy in various settings. We note that our work focuses on the angle of expressivity, whereas the work below focuses on the sample efficiency.

**Tabular MDPs.** The extensive study in tabular MDP setting leaves little gap in their sample complexity of model-based and model-free algorithms, whereas the space complexity seems to be the main difference. (Strehl et al., 2006). The best sample complexity bounds for model-based tabular RL (Azar et al., 2017; Zanette & Brunskill, 2019) and model-free tabular RL (Jin et al., 2018) only differ by a $\text{poly}(H)$ multiplicative factor (where $H$ is the horizon.)

**Linear Quadratic Regulator.** Dean et al. (2018) and Dean et al. (2017) provided sample complexity bound for model-based LQR. Recently, Tu & Recht (2018) analyzed sample efficiency of the model-based and model-free problem in the setting of Linear Quadratic Regulator, and proved an $O(d)$ gap in sample complexity, where $d$ is the dimension of state space. Unlike tabular MDP case, the space complexity of model-based and model-free algorithms has little difference. The sample-efficiency gap mostly comes from that dynamics learning has $d$-dimensional supervisions, whereas $Q$-learning has only one-dimensional supervision.

**Contextual Decision Process (with function approximator).** Sun et al. (2019) prove an exponential information-theoretical gap between mode-based and model-free algorithms in the factored MDP setting. Their definition of model-free algorithms requires an exact parameterization: the value-function hypothesis class should be exactly the family of optimal value-functions induced by the MDP family. This limits the application to deep reinforcement learning where overparameterized neural networks are frequently used. Moreover, a crucial reason for the failure of the model-free algorithms is that the reward is designed to be sparse.

**Related Empirical Work.** A large family of model-based RL algorithms uses existing model-free algorithms of its variant on the learned dynamics. MBPO (Janner et al., 2019), STEVE (Buckman et al., 2018), and MVE (Feinberg et al., 2018) are model-based $Q$-learning-based policy optimization algorithms, which can be viewed as modern extensions and improvements of the early model-based $Q$-learning framework, Dyna (Sutton, 1990). SLBO (Luo et al., 2019) is a model-based policy optimization algorithm using TRPO as the algorithm in the learned environment.

Another way to exploit the dynamics is to use it to perform model-based planning. Racanière et al. (2017) and Du & Narasimhan (2019) use the model to generated additional extra data to do planning implicitly. Chua et al. (2018) study how to combine an ensemble of probabilistic models and planning, which is followed by Wang & Ba (2019), which introduces a policy network to distill knowledge from a planner and provides a prior for the planner. Piché et al. (2018) uses methods in Sequential Monte Carlo in the context of control as inference. Oh et al. (2017) trains a $Q$-function and then perform lookahead planning. Nagabandi et al. (2018) uses random shooting as the planning algorithm. Lowrey et al. (2018) uses the dynamics to improve the performance of model-free algorithms.

Heess et al. (2015) backprops through a stochastic computation graph with a stochastic gradient to optimize the policy under the learned dynamics. Levine & Koltun (2013) distills a policy from trajectory optimization. Rajeswaran et al. (2016) trains a policy adversarially robust to the worst dynamics in the ensemble. Clavera et al. (2018) reformulates the problem as a meta-learning problem and using meta-learning algorithms. Predictron (Silver et al., 2017) learns a dynamics and value function and then use them to predict the future reward sequences.

Another line of work focus on how to improve the learned dynamics model. Many of them use an ensemble of models (Kurutach et al., 2018; Rajeswaran et al., 2016; Clavera et al., 2018), which are further extended to an ensemble of probabilistic models (Chua et al., 2018; Wang & Ba, 2019). Luo et al. (2019) designs a discrepancy bound for learning the dynamics model. Talvitie (2014) augments the data for model training in a way that the model can output a real observation from its own prediction. Malik et al. (2019) calibrates the model's uncertainty so that the model's output distribution should match the frequency of predicted states. Oh et al. (2017) learns a representation of states by predicting rewards and future returns using representation.

## 3    PRELIMINARIES

**Markov Decision Process.** A Markov Decision Process (MDP) is a tuple $\langle \mathcal{S}, \mathcal{A}, f, r, \gamma \rangle$, where $\mathcal{S}$ is the state space, $\mathcal{A}$ the action space, $f : \mathcal{S} \times \mathcal{A} \to \Delta(\mathcal{S})$ the transition dynamics that maps a state action pair to a probability distribution of the next state, $\gamma$ the discount factor, and $r \in \mathbb{R}^{\mathcal{S} \times \mathcal{A}}$ the reward function. Throughout this paper, we will consider deterministic dynamics, which, with slight abuse of notation, will be denoted by $f : \mathcal{S} \times \mathcal{A} \to \mathcal{S}$.

A deterministic policy $\pi : \mathcal{S} \to \mathcal{A}$ maps a state to an action. The value function for the policy is defined as is defined $V^\pi(s) \stackrel{\text{def}}{=} \sum_{h=1}^{\infty} \gamma^{h-1} r(s_h, a_h)$. where $a_h = \pi(s_h), s_1 = s$ and $s_{h+1} = f(s_h, a_h)$.

An RL agent aims to find a policy $\pi$ that maximizes the expected total reward defined as

$$\eta(\pi) \stackrel{\text{def}}{=} \mathbb{E}_{s_1 \sim \mu} \left[ V^\pi(s_1) \right],$$

where $\mu$ is the distribution of the initial state.

**Bellman Equation.** Let $\pi^\star$ be the optimal policy, and $V^\star$ the optimal value function (that is, the value function for policy $\pi^\star$). The value function $V^\pi$ for policy $\pi$ and optimal value function $V^\star$ satisfy the Bellman equation and Bellman optimality equation, respectively. Let $Q^\pi$ and $Q^\star$ defines the state-action value function for policy $\pi$ and optimal state-action value function. Then, for a deterministic dynamics $f$, we have

$$\begin{cases} V^\pi(s) = Q^\pi(s, \pi(s)), \\ Q^\pi(s, a) = r(s, a) + \gamma V^\pi(f(s, a)), \end{cases} \quad \text{and} \quad \begin{cases} V^\star(s) = \max_{a \in \mathcal{A}} Q^\star(s, a), \\ Q^\star(s, a) = r(s, a) + \gamma V^\star(f(s, a)). \end{cases} \quad (1)$$

Denote the Bellman operator for dynamics $f$ by $\mathcal{B}_f$: $(\mathcal{B}_f[Q])(s, a) = r(s, a) + \max_{a'} \gamma Q(f(s, a), a')$.

**Neural Networks.** We focus on fully-connected neural nets with ReLU function as activations. A one-dimensional input and one-dimensional output ReLU neural net represents a piecewise linear function. A two-layer ReLU neural net with $d$ hidden neurons represents a piecewise linear function with at most $(d + 1)$ pieces. Similarly, an $H$-layer neural net with $d$ hidden neurons in each layer represents a piecewise linear function with at most $(d + 1)^H$ pieces (Pascanu et al., 2013).

**Problem Setting and Notations.** In this paper, we focus on continuous state space, discrete action space MDPs with $\mathcal{S} \subset \mathbb{R}$. We assume the dynamics is deterministic (that is, $s_{t+1} = f(s_t, a_t)$), and the reward is known to the agent. Let $\lfloor x \rfloor$ denote the floor function of $x$, that is, the greatest integer less than or equal to $x$. We use $\mathbb{I}[\cdot]$ to denote the indicator function.

## 4    APPROXIMABILITY OF $Q$-FUNCTIONS AND DYNAMICS

We show that there exist MDPs in one-dimensional continuous state space that have simple dynamics but complex $Q$-functions and policies. Moreover, any polynomial-size neural network function approximator of the $Q$-function or policy will result in a sub-optimal expected total reward, and learning $Q$-functions parameterized by neural networks requires fundamentally an exponential number of samples (Section 4.2). Section 4.3 illustrates the phenomena that $Q$-function is more complex than the dynamics occurring frequently and naturally even with random MDP, beyond the theoretical construction.

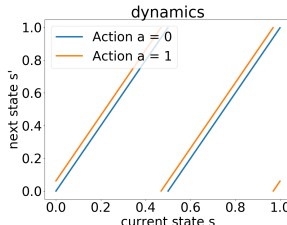 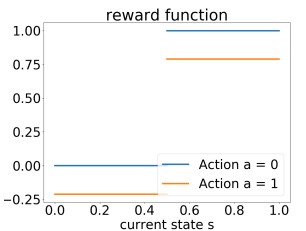 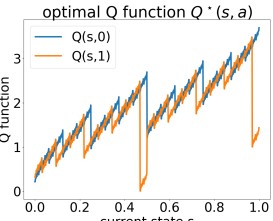

(a) Visualization of dynamics for action $a = 0, 1$.

(b) The reward function $r(s, 0)$ and $r(s, 1)$.

(c) Approximation of optimal $Q$-function $Q^\star(s, a)$

Figure 2: A visualization of the dynamics, the reward function in the MDP defined in Definition 4.1, and the approximation of its optimal $Q$-function for the effective horizon $H = 4$. We can also construct slightly more involved construction with Lipschitz dynamics and very similar properties. Please see Appendix C.

### 4.1 A Provable Construction of MDPs with Complex $Q$

Recall that we consider the infinite horizon case and $0 < \gamma < 1$ is the discount factor. Let $H = (1 - \gamma)^{-1}$ be the "effective horizon" — the rewards after $\gg H$ steps become negligible due to the discount factor. For simplicity, we assume that $H > 3$ and it is an integer. (Otherwise we take just take $H = \lfloor (1 - \gamma)^{-1} \rfloor$.) Throughout this section, we assume that the state space $\mathcal{S} = [0, 1)$ and the action space $\mathcal{A} = \{0, 1\}$.

**Definition 4.1.** *Given the effective horizon $H = (1 - \gamma)^{-1}$, we define an MDP $M_H$ as follows. Let $\kappa = 2^{-H}$. The dynamics $f$ by the following piecewise linear functions with at most three pieces.*

$$f(s, 0) = \begin{cases} 2s & \text{if } s < 1/2 \\ 2s - 1 & \text{if } s \geq 1/2 \end{cases} \qquad f(s, 1) = \begin{cases} 2s + \kappa & \text{if } s < (1 - \kappa)/2 \\ 2s + \kappa - 1 & \text{if } (1 - \kappa)/2 \leq s \leq (2 - \kappa)/2 \\ 2s + \kappa - 2 & \text{otherwise.} \end{cases}$$

*The reward function is defined as*

$$r(s, 0) = \mathbb{I}[1/2 \leq s < 1]$$
$$r(s, 1) = \mathbb{I}[1/2 \leq s < 1] - 2(\gamma^{H-1} - \gamma^H)$$

*The initial state distribution $\mu$ is uniform distribution over the state space $[0, 1)$.*

The dynamics and the reward function for $H = 4$ are visualized in Figures 2a, 2b. Note that by the definition, the transition function for a fixed action $a$ is a piecewise linear function with at most 3 pieces. Our construction can be modified so that the dynamics is Lipschitz and the same conclusion holds (see Appendix C).

Attentive readers may also realize that the dynamics can be also be written succinctly as $f(s, 0) = 2s \mod 1$ and $f(s, 1) = 2s + \kappa \mod 1^2$, which are key properties that we use in the proof of Theorem 4.2 below.

**Optimal $Q$-function $Q^\star$ and the optimal policy $\pi^\star$.** Even though the dynamics of the MDP constructed in Definition 4.1 has only a constant number of pieces, the $Q$-function and policy are very complex: (1) the policy is a piecewise linear function with exponentially number of pieces, (2) the optimal $Q$-function $Q^\star$ and the optimal value function $V^\star$ are actually *fractals* that are not continuous anywhere. These are formalized in the theorem below.

**Theorem 4.2.** *For $s \in [0, 1)$, let $s^{(k)}$ denotes the $k$-th bit of $s$ in the binary representation.[3] The optimal policy $\pi^\star$ for the MDP defined in Definition 4.1 has $2^{H+1}$ number of pieces. In particular,*

$$\pi^\star(s) = \mathbb{I}[s^{(H+1)} = 0]. \tag{2}$$

---

[2] The mod function is defined as: $x \mod 1 \triangleq x - \lfloor x \rfloor$. More generally, for positive real $k$, we define $x \mod k \triangleq x - k \lfloor x/k \rfloor$.

[3] Or more precisely, we define $s^{(h)} \triangleq \lfloor 2^h s \rfloor \mod 2$.

*And the optimal value function is a fractal with the expression:*

$$V^\star(s) = \sum_{h=1}^{H} \gamma^{h-1} s^{(h)} + \sum_{h=H+1}^{\infty} \gamma^{h-1}\left(1 + 2(s^{(h+1)} - s^{(h)})\right) + \gamma^{H-1}\left(2s^{(H+1)} - 2\right). \quad (3)$$

*The close-form expression of $Q^\star$ can be computed by $Q^\star(s,a) = r(s,a) + V^\star(f(s,a))$, which is also a fractal.*

We approximate the optimal $Q$-function by truncating the infinite sum to $2H$ terms, and visualize it in Figure 2c. We discuss the main intuitions behind the construction in the following proof sketch of the Theorem. A rigorous proof of Theorem 4.2) is deferred to Appendix B.1.

*Proof Sketch.* The key observation is that the dynamics $f$ essentially shift the binary representation of the states with some addition. We can verify that the dynamics satisfies $f(s,0) = 2s \mod 1$ and $f(s,1) = 2s + \kappa \mod 1$ where $\kappa = 2^{-H}$. In other words, suppose $s = 0.s^{(1)}s^{(2)}\cdots$ is the binary representation of $s$, and let left-shift$(s) = 0.s^{(2)}s^{(3)}\cdots$.

$$f(s,0) = \text{left-shift}(s) \quad (4)$$
$$f(s,1) = (\text{left-shift}(s) + 2^{-H}) \mod 1 \quad (5)$$

Moreover, the reward function is approximately equal to the first bit of the binary representation

$$r(s,0) = s^{(1)}, \quad r(s,a) \approx s^{(1)} \quad (6)$$

(Here the small negative drift of reward for action $a = 1$, $-2(\gamma^{H-1} - \gamma^H)$, is only mostly designed for the convenience of the proof, and casual readers can ignore it for simplicity.) Ignoring carries, the policy pretty much can only affect the $H$-th bit of the next state $s' = f(s,a)$: the $H$-th bit of $s'$ is either equal to $(H+1)$-th bit of $s$ when action is 0, or equal its flip when action is 1. Because the bits will eventually be shifted left and the reward is higher if the first bit of a future state is 1, towards getting higher future reward, the policy should aim to create more 1's. Therefore, the optimal policy should choose action 0 if the $(H+1)$-th bit of $s$ is already 1, and otherwise choose to flip the $(H+1)$-th bit by taking action 1.

A more delicate calculation that addresses the carries properly would lead us to the form of the optimal policy (Equation (2).) Computing the total reward by executing the optimal policy will lead us to the form of the optimal value function (equation (3).) (This step does require some elementary but sophisticated algebraic manipulation.)

With the form of the $V^\star$, a shortcut to a formal, rigorous proof would be to verify that it satisfies the Bellman equation, and verify $\pi^\star$ is consistent with it. We follow this route in the formal proof of Theorem 4.2) in Appendix B.1. $\square$

### 4.2 THE APPROXIMABILITY OF $Q$-FUNCTION

A priori, the complexity of $Q^\star$ or $\pi^\star$ does not rule out the possibility that there exists an approximation of them that do an equally good job in terms of maximizing the rewards. However, we show that in this section, indeed, there is no neural network approximation of $Q^\star$ or $\pi^\star$ with a polynomial width. We prove this by showing any piecewise linear function with a sub-exponential number of pieces cannot approximate either $Q^\star$ or $\pi^\star$ with a near-optimal total reward.

**Theorem 4.3.** *Let $M_H$ be the MDP constructed in Definition 4.1. Suppose a piecewise linear policy $\pi$ has a near optimal reward in the sense that $\eta(\pi) \geq 0.92 \cdot \eta(\pi^\star)$, then it has to have at least $\Omega\left(\exp(cH)/H\right)$ pieces for some universal constant $c > 0$. As a corollary, no constant depth neural networks with polynomial width (in $H$) can approximate the optimal policy with near optimal rewards.*

Consider a policy $\pi$ induced by a value function $Q$, that is, $\pi(s) = \arg\max_{a \in \mathcal{A}} Q(s,a)$. Then, when there are two actions, the number of pieces of the policy is bounded by twice the number of pieces of $Q$. This observation and the theorem above implies the following inapproximability result of $Q^\star$.

**Corollary 4.4.** *In the setting of Theorem 4.3, let $\pi$ be the policy induced by some $Q$. If $\pi$ is near-optimal in a sense that $\eta(\pi) \geq 0.92 \cdot \eta(\pi^\star)$, then $Q$ has at least $\Omega\left(\exp(cH)/H\right)$ pieces for some universal constant $c > 0$.*

The intuition behind the proof of Theorem 4.3 is as follows. Recall that the optimal policy has the form $\pi^\star(s) = \mathbb{I}[s^{(H+1)} = 0]$. One can expect that any polynomial-pieces policy $\pi$ behaves suboptimally in most of the states, which leads to the suboptimality of $\pi$. Detailed proof of Theorem 4.3 is deferred to Appendix B.2.

Beyond the expressivity lower bound, we also provide an exponential sample complexity lower bound for Q-learning algorithms parameterized with neural networks (see Appendix B.4).

## 4.3 THE APPROXIMABILITY OF $Q$-FUNCTIONS OF RANDOMLY GENERATED MDPS

In this section, we show the phenomena that the $Q$-function not only occurs in the crafted cases as in the previous subsection, but also occurs more robustly with a decent chance for (semi-) randomly generated MDPs. (Mathematically, this says that the family of MDPs with such a property is not a degenerate measure-zero set.)

It is challenging and perhaps requires deep math to characterize the fractal structure of $Q$-functions for random dynamics, which is beyond the scope of this paper. Instead, we take an empirical approach here. We generate random piecewise linear and Lipschitz dynamics, and compute their $Q$-functions for the finite horizon, and then visualize the $Q$-functions or count the number of pieces in the $Q$-functions. We also use DQN algorithm (Mnih et al., 2015) with a finite-size neural network to learn the $Q$-function.

We set horizon $H = 10$ for simplicity and computational feasibility. The state and action space are $[0, 1)$ and $\{0, 1\}$ respectively. We design two methods to generate random or semi-random piecewise dynamics with at most four pieces. First, we have a uniformly random method, called RAND, where we independently generate two piecewise linear functions for $f(s, 0)$ and $f(s, 1)$, by generating random positions for the kinks, generating random outputs for the kinks, and connecting the kinks by linear lines (See Appendix D.1 for a detailed description.)

In the second method, called SEMI-RAND, we introduce a bit more structure in the generation process, towards increasing the chance to see the phenomenon. The functions $f(s, 0)$ and $f(s, 1)$ have 3 pieces with shared kinks. We also design the generating process of the outputs at the kinks so that the functions have more fluctuations. The reward for both of the two methods is $r(s, a) = s, \forall a \in \mathcal{A}$. (See Appendix D.1 for a detailed description.)

Figure 1 illustrates the dynamics of the generated MDPs from SEMI-RAND. More details of empirical settings can be found in Appendix D.1. **The optimal policy and $Q$ can have a large number of pieces.** Because the state space has one dimension, and the horizon is 10, we can compute the exact $Q$-functions by recursively applying Bellman operators, and count the number of pieces. We found that, $8.6\%$ fraction of the 1000 MDPs independently generated from the RAND method has policies with more than 100 pieces, much larger than the number of pieces in the dynamics (which is 4). Using the SEMI-RAND method, a $68.7\%$ fraction of the MDPs has polices with more than $10^3$ pieces. In Section D.1, we plot the histogram of the number of pieces of the $Q$-functions. Figure 1 visualize the $Q$-functions and dynamics of two MDPs generated from RAND and SEMI-RAND method. These results suggest that the phenomenon that $Q$-function is more complex than dynamics is not a degenerate phenomenon and can occur with non-zero measure. For more empirical results, see Appendix D.2.

**Model-based policy optimization methods also suffer from a lack of expressivity.** As an implication of our theory in the previous section, when the $Q$-function or the policy are too complex to be approximated by a reasonable size neural network, both model-free algorithms or model-based policy optimization algorithms will suffer from the lack of expressivity, and as a consequence, the sub-optimal rewards. We verify this claim on the randomly generated MDPs discussed in Section 4.3, by running DQN (Mnih et al., 2015), SLBO (Luo et al., 2019), and MBPO (Janner et al., 2019) with various architecture size.

For the ease of exposition, we use the MDP visualized in the bottom half of Figure 1. The optimal policy for this specific MDP has 765 pieces, and the optimal $Q$-function has about $4 \times 10^4$ number of pieces, and we can compute the optimal total rewards.

First, we apply DQN to this environment by using a two-layer neural network with various widths to parameterize the $Q$-function. The training curve is shown in Figure 3 (Left). Model-free algorithms

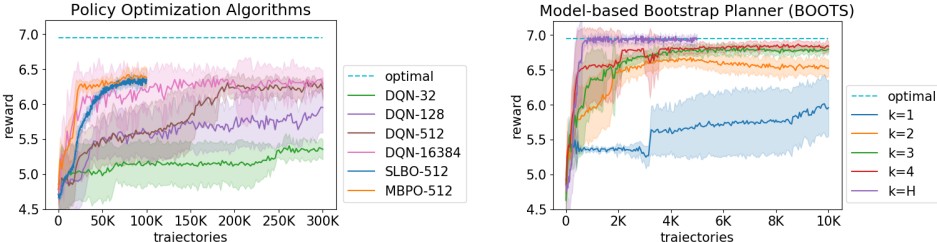

Figure 3: **(Left):** The performance of DQN, SLBO, and MBPO on the bottom dynamics in Figure 1. The number after the acronym is the width of the neural network used in the parameterization of $Q$. We see that even with sufficiently large neural networks and sufficiently many steps, these algorithms still suffers from bad approximability and cannot achieve optimal reward. **(Right):** Performance of BOOTS + DQN with various planning steps. A near-optimal reward is achieved with even $k = 3$, indicating that the bootstrapping with the learned dynamics improves the expressivity of the policy significantly.

---

**Algorithm 1** Model-based Bootstrapping Planner (BOOTS) + RL Algorithm X

---

1: **training:** run Algorithm X, store the all samples in the set $R$, store the learned $Q$-function $Q$, and the learned dynamics $\hat{f}$ if it is available in Algorithm X.
2: **testing:**
3:     if $\hat{f}$ is not available, learn $\hat{f}$ from the data in $R$
4:     execute the policy BOOTS(s) at every state $s$
5:
1: **function** BOOTS(s)
2:     **Given:** query oracle for function $Q$ and $\hat{f}$
3:     Compute

$$\pi^{\text{boots}}_{k,Q,\hat{f}}(s) = \arg\max_a \max_{a_1,\dots,a_k} r(s,a) + \cdots + \gamma^{k-1} r(s_{k-1}, a_{k-1}) + \gamma^k Q(s_k, a_k)$$

using a zero-th order optimization algorithm (which only requires oracle query of the function value) such as cross-entropy method or random shooting.

---

can not find near-optimal policy even with $2^{14}$ hidden neurons and 1M trajectories, which suggests that there is a fundamental approximation issue. This result is consistent with Fu et al. (2019), in a sense that enlarging Q-network improves the performance of DQN algorithm at convergence.

Second, we apply SLBO and MBPO in the same environment. Because the policy network and $Q$-function in SLOBO and MBPO cannot approximate the optimal policy and value function, we see that they fail to achieve near-optimal rewards, as shown in Figure 3 (Left).

## 5 MODEL-BASED BOOTSTRAPPING PLANNER

Our theory and experiments in Section 4.2 and 4.3 demonstrate that when the $Q$-function or the policy is complex, model-free or model-based policy optimization algorithms will suffer from the lack of expressivity. The intuition suggests that model-based planning algorithms will not suffer from the lack of expressivity because the final policy is not represented by a neural network. For the construction in Section 4.1, we can actually prove that even a few-steps planner can bootstrap the expressivity of the $Q$-function (formalized in Theorem 5.1 below).

Inspired the theoretical result, we apply a simple $k$-step model-based bootstrapping planner on top of existing $Q$-functions (trained from either model-based or model-free approach) *in the test time*, on either the one-dimensional MDPs considered in Section 4 or the continuous control benchmark tasks in MuJoCo. The bootstrapping planner is reminiscent of MCTS using in AlphaGo (Silver et al., 2016; 2018). However, here, we use the learned dynamics and deal with the continuous state space.

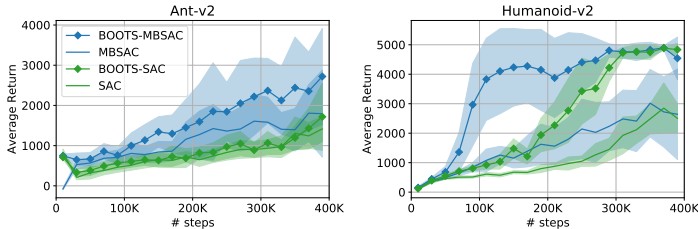

Figure 4: Comparison of BOOTS-MBSAC vs MBSAC and BOOTS-SAC vs SAC on Ant and Humanoid. Particularly on the Humanoid environment, BOOTS improves the performance significantly. The test policies for MBSAC and SAC are the deterministic policy that takes the mean of the output of the policy network, because the deterministic policy performs better than the stochastic policy in the test time.

## 5.1 BOOTSTRAPPING THE $Q$-FUNCTION

Given a function $Q$ that is potentially not expressive enough to approximate the optimal $Q$-function, we can apply the Bellman operator with a learned dynamics $\hat{f}$ for $k$ times to get a bootstrapped version of $Q$:

$$\mathcal{B}_{\hat{f}}^k[Q] = \underbrace{\mathcal{B}_{\hat{f}}[\cdots[\mathcal{B}_{\hat{f}}[Q]]]}_{k \text{ times}} \tag{7}$$

$$\text{or} \quad \mathcal{B}_{\hat{f}}^k[Q](s,a) = \max_{a_1,\cdots,a_k} r(s_0,a_0) + \cdots + \gamma^{k-1} r(s_{k-1},a_{k-1}) + \gamma^k Q(s_k,a_k) \tag{8}$$

where $s_0 = s, a_0 = a$ and $s_{h+1} = \hat{f}(s_h, a_h)$.

Given the bootstrapped version, we can derive a greedy policy w.r.t it:

$$\pi_{k,Q,\hat{f}}^{\text{boots}}(s) = \max_a \mathcal{B}_{\hat{f}}^k[Q](s,a) \tag{9}$$

Algorithm 1, called BOOTS summarizes how to apply the planner on top of any RL algorithm with a $Q$-function (straightforwardly).

For the MDPs constructed in Section 4.1, we can prove that representing the optimal $Q$-function by $\mathcal{B}_{\hat{f}}^k[Q]$ requires fewer pieces in $Q$ than representing the optimal $Q$-function by $Q$ directly.

**Theorem 5.1.** *Consider the MDP $M_H$ defined in Definition 4.1. There exists a constant-piece piecewise linear dynamics $\hat{f}$ and a $2^{H-k+1}$-piece piecewise linear function $Q$, such that the bootstrapped policy $\pi_{k,Q,\hat{f}}^{\text{boots}}(s)$ achieves the optimal total rewards.*

By contrast, recall that in Theorem 4.3, we show that approximating the optimal $Q$-function directly with a piecewise linear function requires $\approx 2^H$ piecewise. Thus we have a multiplicative factor of $2^k$ gain in the expressivity by using the bootstrapped policy. Here the exponential gain is only magnificent enough when $k$ is close to $H$ because the gap of approximability is huge. However, in more realistic settings — the randomly-generated MDPs and the MuJoCo environment — the bootstrapping planner improvs the performance significantly as shown in the next subsection.

## 5.2 EXPERIMENTS

**BOOTS on random piecewise linear MDPs.** We implement BOOTS (Algorithm 1) with various steps of planning and with the learned dynamics.[4] . The planner is an exponential-time planner which enumerates all the possible future sequence of actions. We also implement bootstrapping with partial planner with varying planning horizon. As shown in Figure 3, BOOTS + DQN not only has the best sample-efficiency, but also achieves the optimal reward. In the meantime, even a partial planner helps to improve both the sample-efficiency and performance. More details of this experiment are deferred to Appendix D.3.

---

[4]Our code is available at `https://github.com/roosephu/boots`.

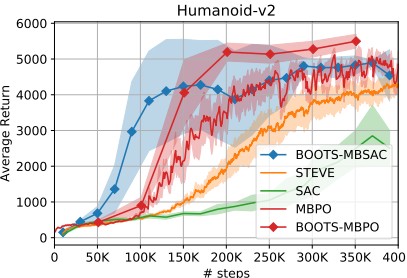

Figure 5: BOOTS-MBSAC or BOOTS-MBPO outperforms previous state-of-the-art algorithms on Humanoid. The results are averaged over 5 random seeds and shadow area indicates a single standard deviation from the mean.

**BOOTS on MuJoCo environments.** We work with the OpenAI Gym environments (Brockman et al., 2016) based on the Mujoco simulator (Todorov et al., 2012) with maximum horizon 1000 and discount factor 1. We apply BOOTS on top of three algorithms: (a) **SAC** (Haarnoja et al., 2018), the state-of-the-art model-free RL algorithm; (b) **MBPO** (Janner et al., 2019), a model-based Q-learning algorithm, and an extension of Dyna (Sutton, 1990); (c) a computationally efficient variant of MBPO that we develop using ideas from SLBO (Luo et al., 2019), which is called **MBSAC**. The main difference here from MBPO and other works such as (Wang & Ba, 2019; Kurutach et al., 2018) is that we don't use model ensemble. Instead, we occasionally optimize the dynamics by one step of Adam to introduce stochasticity in the dynamics, following the technique in SLBO Luo et al. (2019). Our algorithm is a few times faster than MBPO in wall-clock time. It performs similarly to MBPO on Humanoid, but generally worse than MBPO on other environments. See Appendix A.1 for details.

We use $k = 4$ steps of planning unless explicitly mentioned otherwise in the ablation study (Section A.2). In Figure 4, we compare BOOTS+SAC with SAC, and BOOTS + MBSAC with MBSAC on Gym Ant and Humanoid environments, and demonstrate that BOOTS can be used on top of existing strong baselines. We found that BOOTS has little help for other simpler environments, and we suspect that those environments have much less complex $Q$-functions so that our theory and intuitions do not necessarily apply. (See Section A.2 for more ablation study.)

In Figure 5, we compare BOOTS+MBSAC and BOOTS+MBPO with other MBPO, SAC, and STEVE (Buckman et al., 2018)[5] on the humanoid environment. We see a strong performance surpassing the previous state-of-the-art MBPO.

## 6 CONCLUSION

Our study suggests that there exists a significant representation power gap of neural networks between for expressing $Q$-function, the policy, and the dynamics in both constructed examples and empirical benchmarking environments. We show that our model-based bootstrapping planner BOOTS helps to overcome the approximation issue and improves the performance in synthetic settings and in the difficult MuJoCo environments. We raise some interesting open questions.

- Can we theoretically generalize our results to high-dimensional state space, or continuous actions space? Can we theoretically analyze the number of pieces of the optimal $Q$-function of a stochastic dynamics?

- In this paper, we measure the complexity by the size of the neural networks. It's conceivable that for real-life problems, the complexity of a neural network can be better measured by its weights norm. Could we build a more realistic theory with another measure of complexity?

- The BOOTS planner comes with a cost of longer test time. How do we efficiently plan in high-dimensional dynamics with a long planning horizon?

- The dynamics can also be more complex (perhaps in another sense) than the $Q$-function in certain cases. How do we efficiently identify the complexity of the optimal $Q$-function, policy, and the dynamics, and how do we deploy the best algorithms for problems with different characteristics?

---

[5]For STEVE, we use the official code at `https://github.com/tensorflow/models/tree/master/research/steve`

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

# A    EXPERIMENT DETAILS IN SECTION 5.2

## A.1    MODEL-BASED SAC (MBSAC)

Here we describe our MBSAC algorithm in Algorithm 2, which is a model-based policy optimization and is used in BOOTS-MBSAC. As mentioned in Section 5.2, the main difference from MBPO and other works such as (Wang & Ba, 2019; Kurutach et al., 2018) is that we don't use model ensemble. Instead, we occasionally optimize the dynamics by one step of Adam to introduce stochasticity in the dynamics, following the technique in SLBO (Luo et al., 2019). As argued in (Luo et al., 2019), the stochasticity in the dynamics can play a similar role as the model ensemble. Our algorithm is a few times faster than MBPO in wall-clock time. It performs similarly to MBPO on Humanoid, but a bit worse than MBPO in other environments. In MBSAC, we use SAC to optimize the policy $\pi_\beta$ and the $Q$-function $Q_\varphi$. We choose SAC due to its sample-efficiency, simplicity and off-policy nature. We mix the real data from the environment and the virtual data which are always fresh and are generated by our learned dynamics model $\hat{f}_\theta$.[6]

---

**Algorithm 2** MBSAC

---

1: Parameterize the policy $\pi_\beta$, dynamics $\hat{f}_\theta$, and the $Q$-function $Q_\varphi$ by neural networks. Initialize replay buffer $\mathcal{B}$ with $n_{\text{init}}$ steps of interactions with the environments by a random policy, and pretrain the dynamics on the data in the replay buffer.
2: $t \leftarrow 0$, and sample $s_0$ from the initial state distribution.
3: **for** $n_{\text{iter}}$ iterations **do**
4:     Perform action $a_t \sim \pi_\beta(\cdot|s_t)$ in the environment, obtain $s'$ as the next state from the environment.
5:     $s_{t+1} \leftarrow s'$, and add the transition $(s_t, a_t, s_{t+1}, r_t)$ to $\mathcal{B}$.
6:     $t \leftarrow t + 1$. If $t = T$ or the trajectory is done, reset to $t = 0$ and sample $s_0$ from the initial state distribution.
7:     **for** $n_{\text{policy}}$ iterations **do**
8:         **for** $n_{\text{model}}$ iterations **do**
9:             Optimize $\hat{f}_\theta$ with a mini-batch of data from $\mathcal{B}$ by one step of Adam.
10:        Sample $n_{\text{real}}$ data $\mathcal{B}_{\text{real}}$ and $n_{\text{start}}$ data $\mathcal{B}_{\text{start}}$ from $\mathcal{B}$.
11:        Perform $q$ steps of **virtual** rollouts using $\hat{f}_\theta$ and policy $\pi_\beta$ starting from states in $\mathcal{B}_{\text{start}}$; obtain $\mathcal{B}_{\text{virtual}}$.
12:        Update $\pi_\beta$ and $Q_\varphi$ using the mini-batch of data in $\mathcal{B}_{\text{real}} \cup \mathcal{B}_{\text{virtual}}$ by SAC.

---

For Ant, we modify the environment by adding the $x$ and $y$ axis to the observation space to make it possible to compute the reward from observations and actions. For Humanoid, we add the position of center of mass. We don't have any other modifications. All environments have maximum horizon 1000.

For the policy network, we use an MLP with ReLU activation function and two hidden layers, each of which contains 256 hidden units. For the dynamics model, we use a network with 2 Fixup blocks (Zhang et al., 2019), with convolution layers replaced by a fully connected layer. We found out that with similar number of parameters, fixup blocks leads to a more accurate model in terms of validation loss. Each fixup block has 500 hidden units. We follow the model training algorithm in Luo et al. (2019) in which non-squared $\ell_2$ loss is used instead of the standard MSE loss.

## A.2    ABLATION STUDY

**Planning with oracle dynamics and more environments.** We found that BOOTS has smaller improvements on top of MBSAC and SAC for the environment Cheetah and Walker. To diagnose the issue, we also plan with an oracle dynamics (the true dynamics). This tells us whether the lack of improvement comes from inaccurate learned dynamics. The results are presented in two ways in Figure 6 and Figure 7. In Figure 6, we plot the mean rewards and the standard deviation of various methods across the randomness of multiple seeds. However, the randomness from the seeds

---

[6]In the paper of MBPO (Janner et al., 2019), the authors don't explicitly state their usage of real data in SAC; the released code seems to make such use of real data, though.

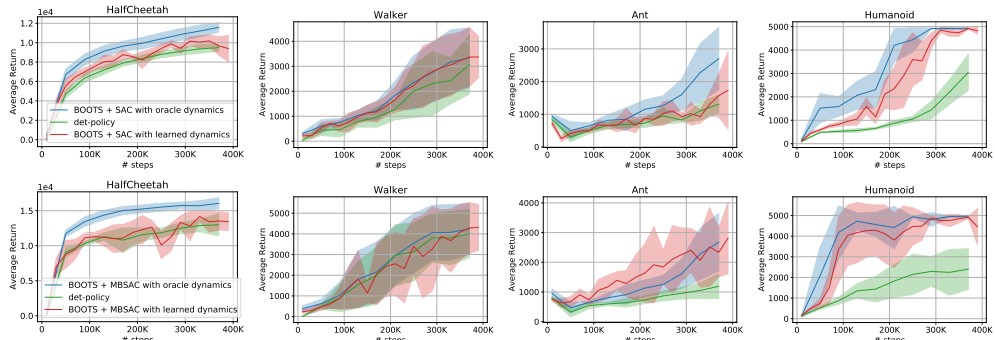

Figure 6: BOOTS with oracle dynamics on top of SAC (top) and MBSAC (bottom) on HalfCheetah, Walker, Ant and Humanoid. The solid lines are average over 5 runs, and the shadow areas indicate the standard deviation.

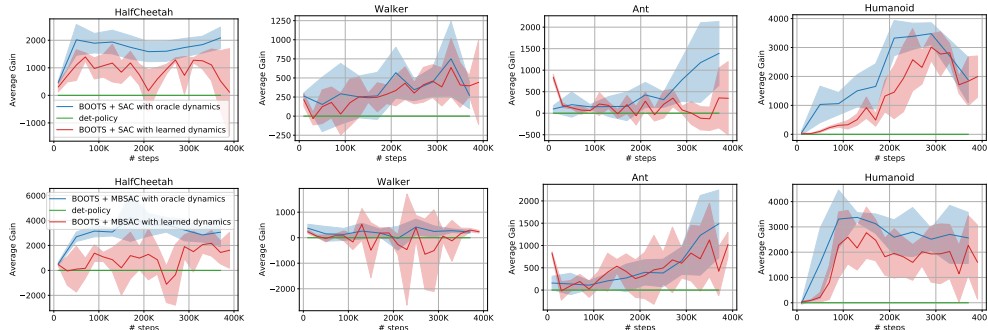

Figure 7: The relative gains of BOOTS over SAC (top) and MBSAC (bottom) on HalfCheetah, Walker, Ant and Humanoid. The solid lines are average over 5 runs, and the shadow areas indicate the standard deviation.

somewhat obscures the gains of BOOTS on each individual run. Therefore, for completeness, we also plot the relative gain of BOOTS on top of MBSAC and SAC, and the standard deviation of the gains in Figure 7.

From Figure 7 we can see planning with the oracle dynamics improves the performance in most of the cases (but with various amount of improvements). However, the learned dynamics sometimes not always can give an improvement similar to the oracle dynamics. This suggests the learned dynamics is not perfect, but oftentimes can lead to good planning. This suggests the expressivity of the $Q$-functions varies depending on the particular environment. How and when to learn and use a learned dynamics for planning is a very interesting future open question.

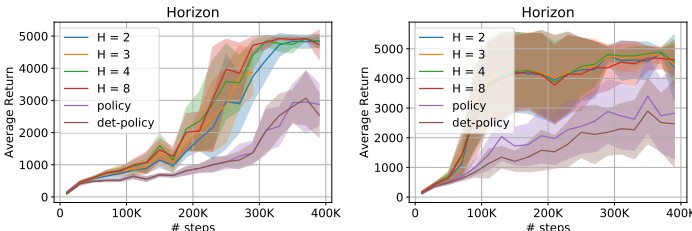

Figure 8: Different BOOTS planning horizon $k$ on top of SAC (left) and MBSAC (right) on Humanoid. The solid lines are average over 5 runs, and the shadow areas indicate the standard deviation.

**The effect of planning horizon.** We experimented with different planning horizons in Figure 8. By planning with a longer horizon, we can earn slightly higher total rewards for both MBSAC and SAC. Planning horizon $k = 16$, however, does not work well. We suspect that it's caused by the compounding effect of the errors in the dynamics.

# B    OMITTED PROOFS IN SECTION 4

In this section we provide the proofs omitted in Section 4.

## B.1    PROOF OF THEOREM 4.2

*Proof of Theorem 4.2.* Since the solution to Bellman optimal equations is unique, we only need to verify that $V^\star$ and $\pi^\star$ defined in equation (1) satisfy the following,

$$V^\star(s) = r(s, \pi^\star(s)) + \gamma V^\star(f(s, \pi^\star(s))), \tag{10}$$
$$V^\star(s) \geq r(s, a) + \gamma V^\star(f(s, a)), \quad \forall a \neq \pi^\star(s). \tag{11}$$

Recall that $s^{(i)}$ is the $i$-th bit in the binary representation of $s$, that is, $s^{(i)} = \lfloor 2^i s \rfloor \mod 2$. Let $\hat{s} = f(s, \pi^\star(s))$. Since $\pi^\star(s) = \mathbb{I}[s^{(H+1)} = 0]$, which ensures the $H$-bit of the next state is 1, we have

$$\hat{s}^{(i)} = \begin{cases} s^{(i+1)}, & i \neq H, \\ 1, & i = H. \end{cases} \tag{12}$$

For simplicity, define $\varepsilon = 2(\gamma^{H-1} - \gamma^H)$. The definition of $r(s, a)$ implies that

$$r(s, \pi^\star(s)) = \mathbb{I}[1/2 \leq s < 1] - \mathbb{I}[\pi^\star(s) = 1]\varepsilon = s^{(1)} - \left(1 - s^{(H+1)}\right)\varepsilon.$$

By elementary manipulation, Eq. (3) is equivalent to

$$V^\star(s) = \sum_{i=1}^{H} \gamma^{i-1} s^{(i)} + \sum_{i=H+1}^{\infty} \left(\gamma^{i-1} - 2(\gamma^{i-2} - \gamma^{i-1})\left(1 - s^{(i)}\right)\right), \tag{13}$$

Now, we verify Eq. (10) by plugging in the proposed solution (namely, Eq. (13)). As a result,

$$r(s, \pi^\star(s)) + \gamma V^\star(\hat{s})$$
$$= s^{(1)} - \left(1 - s^{(H+1)}\right)\varepsilon + \gamma \sum_{i=1}^{H} \gamma^{i-1}\mathbb{I}[\hat{s}^{(i)} = 1] + \gamma \sum_{i=H+1}^{\infty} \left(\gamma^{i-1} - \left(1 - \hat{s}^{(i)}\right)2(\gamma^{i-2} - \gamma^{i-1})\right)$$
$$= s^{(1)} - \left(1 - s^{(H+1)}\right)\varepsilon + \sum_{i=2}^{H} \gamma^{i-1}s^{(i)} + \gamma^H + \sum_{i=H+2}^{\infty} \left(\gamma^{i-1} - \left(1 - s^{(i)}\right)2(\gamma^{i-2} - \gamma^{i-1})\right)$$
$$= \sum_{i=1}^{H} \gamma^{i-1}s^{(i)} + \sum_{i=H+1}^{\infty} \left(\gamma^{i-1} - \left(1 - s^{(i)}\right)2(\gamma^{i-2} - \gamma^{i-1})\right)$$
$$= V^\star(s),$$

which verifies Eq. (10).

In the following we verify Eq. (11). Consider any $a \neq \pi^\star(s)$. Let $\bar{s} = f(s, a)$ for shorthand. Note that $\bar{s}^{(i)} = s^{(i+1)}$ for $i > H$. As a result,

$$
\begin{aligned}
& V^\star(s) - \gamma V^\star(\bar{s}) \\
&= \sum_{i=1}^{H} \gamma^{i-1} s^{(i)} + \sum_{i=H+1}^{\infty} \left( \gamma^{i-1} - \left(1 - s^{(i)}\right) 2(\gamma^{i-2} - \gamma^{i-1}) \right) \\
&\quad - \sum_{i=1}^{H} \gamma^{i-1} \bar{s}^{(i)} - \sum_{i=H+1}^{\infty} \left( \gamma^{i-1} - \left(1 - \bar{s}^{(i)}\right) 2(\gamma^{i-2} - \gamma^{i-1}) \right) \\
&= s^{(1)} + \sum_{i=1}^{H-1} \gamma^i \left( s^{(i+1)} - \bar{s}^{(i)} \right) - \gamma^H \bar{s}^{(H)} + \gamma^H - 2 \left( 1 - s^{(H+1)} \right) \left( \gamma^{H-1} - \gamma^H \right)
\end{aligned}
$$

For the case where $s^{(H+1)} = 0$, we have $\pi^\star(s) = 1$. For $a = 0$, $\bar{s}^{(i)} = s^{(i+1)}$ for all $i \geq 1$. Consequently,

$$
V^\star(s) - \gamma V^\star(\bar{s}) = s^{(1)} + \gamma^H - \varepsilon > s^{(1)} = r(s, 0),
$$

where the last inequality holds when $\gamma^H - \varepsilon > 0$, or equivalently, $\gamma > 2/3$.

For the case where $s^{(H+1)} = 1$, we have $\pi^\star(s) = 0$. For $a = 1$, we have $s^{(H+1)} = 1$ and $\bar{s}^{(H)} = 0$. Let $p = \max\{i \leq H : s^{(i)} = 0\}$, where we define the max of an empty set is 0. The dynamics $f(s, 1)$ implies that

$$
\bar{s}^{(i)} = \begin{cases} s^{(i+1)}, & i+1 < p \text{ or } i > H, \\ 1, & i+1 = p, \\ 0, & p < i+1 \leq H+1. \end{cases}
$$

Therefore,

$$
V^\star(s) - \gamma V^\star(\bar{s}) = s^{(1)} + \gamma^H + \sum_{i=1}^{H-1} \gamma^i \left( s^{(i+1)} - \bar{s}^{(i)} \right) > s^{(1)} - \varepsilon = r(s, 1).
$$

In both cases, we have $V^\star - \gamma V^\star(\bar{s}) > r(s, a)$ for $a \neq \pi^\star(s)$, which proves Eq. (11). $\qquad \square$

## B.2 PROOF OF THEOREM 4.3

For a fixed parameter $H$, let $z(\pi)$ be the number of pieces in $\pi$. For a policy $\pi$, define the state distribution when acting policy $\pi$ at step $h$ as $\mu_h^\pi$.

In order to prove Theorem 4.3, we show that if $1/2 - 2Hz(\pi)/2^H < 0.3$, then $\eta(\pi) < 0.92\eta(\pi^\star)$. The proof is based on the advantage decomposition lemma.

**Lemma B.1** (Advantage Decomposition Lemma (Schulman et al., 2015; Kakade & Langford, 2002)). *Define* $A^\pi(s, a) = r(s, a) + \gamma V^\pi(f(s, a)) - V^\pi(s) = Q^\pi(s, a) - V^\pi(s)$. *Given policies $\pi$ and $\tilde{\pi}$, we have*

$$
\eta(\pi) = \eta(\tilde{\pi}) + \sum_{h=1}^{\infty} \gamma^{h-1} \mathbb{E}_{s \sim \mu_h^\pi} \left[ A^{\tilde{\pi}}(s, \pi(s)) \right]. \tag{14}
$$

**Corollary B.2.** *For any policy $\pi$, we have*

$$
\eta(\pi^\star) - \eta(\pi) = \sum_{h=1}^{\infty} \gamma^{h-1} \mathbb{E}_{s \sim \mu_h^\pi} \left[ V^\star(s) - Q^\star(s, \pi(s)) \right]. \tag{15}
$$

Intuitively speaking, since $\pi^\star = \mathbb{I}[s^{(H+1)} = 0]$, the a policy $\pi$ with polynomial pieces behaves suboptimally in most of the states. Lemma B.3 shows that the single-step suboptimality gap $V^\star(s) - Q^\star(s, \pi(s))$ is large for a constant portion of the states. On the other hand, Lemma B.4 proves that

the state distribution $\mu_h^\pi$ is near uniform, which means that suboptimal states can not be avoided. Combining with Corollary B.2, the suboptimal gap of policy $\pi$ is large.

The next lemma shows that, if $\pi$ does not change its action for states from a certain interval, the average advantage term $V^\star(s) - Q^\star(s, \pi(s))$ in this interval is large. Proof of this lemma is deferred of Section B.3.

**Lemma B.3.** *Let $\ell_k = [k/2^H, (k+1)/2^H)$, and $\mathcal{K} = \{0 \le k < 2^H : k \mod 2 = 1\}$. Then for $k \in \mathcal{K}$, if policy $\pi$ does not change its action at interval $\ell_k$ (that is, $|\{\pi(s) : s \in \ell_k\}| = 1$), we have*

$$\frac{1}{|\ell_k|} \int_{s \in \ell_k} (V^\star(s) - Q^\star(s, \pi(s)))\, ds \ge 0.183 \tag{16}$$

*for $H > 500$.*

Next lemma shows that when the number of pieces in $\pi$ is not too large, the distribution $\mu_h^\pi$ is close to uniform distribution for step $1 \le h \le H$. Proof of this lemma is deferred of Section B.3

**Lemma B.4.** *Let $z(\pi)$ be the number of pieces of policy $\pi$. For $k \in [2^H]$, define interval $\ell_k = [k/2^H, (k+1)/2^H)$. Let $\nu_h(k) = \inf_{s \in \ell_k} \mu_h^\pi(s)$, If the initial state distribution $\mu$ is uniform distribution, then for any $h \ge 1$,*

$$\sum_{0 \le k < 2^H} 2^{-H} \cdot \nu_h(k) \ge 1 - 2h\frac{z(\pi)}{2^H}. \tag{17}$$

Now we present the proof for Theorem 4.3.

*Proof of Theorem 4.3.* For any $k \in [2^H]$, consider the interval $\ell_k = [k/2^H, (k+1)/2^H)$. Let $\mathcal{K} = \{k \in [A^H] : k \mod 2 = 1\}$. If $\pi$ does not change at interval $\ell_k$ (that is, $|\{\pi(s) : s \in \ell_k\}| = 1$), by Lemma B.3 we have

$$\int_{s \in \ell_k} (V^\star(s) - Q^\star(s, \pi(s)))\, ds \ge 0.183 \cdot 2^{-H}. \tag{18}$$

Let $\nu_h(k) = \inf_{s \in \ell_k} \mu_h^\pi(s)$, then by advantage decomposition lemma (namely, Corollary B.2), we have

$$\begin{aligned}
\eta(\pi^\star) - \eta(\pi) &= \sum_{h=1}^{\infty} \gamma^{h-1} \left( \int_{s \in [0,1)} (V^*(s) - Q^\star(s, \pi(s)))\, d\mu_h^\pi(s) \right) \\
&\ge \sum_{h=1}^{10H} \gamma^{h-1} \left( \sum_{k \in \mathcal{K}} \int_{s \in \ell_k} (V^*(s) - Q^\star(s, \pi(s)))\, d\mu_h^\pi(s) \right) \\
&\ge \sum_{h=1}^{10H} \gamma^{h-1} \left( \sum_{k \in \mathcal{K}} \int_{s \in \ell_k} \nu_h(k)(V^*(s) - Q^\star(s, \pi(s)))\, ds \right) \\
&\ge \sum_{h=1}^{10H} \gamma^{h-1} \left( \sum_{k \in \mathcal{K}} 0.183 \cdot 2^{-H} \cdot \nu_h(k) \right).
\end{aligned}$$

By Lemma B.4 and union bound, we get

$$\sum_{k \in \mathcal{K}} 2^{-H} \cdot \nu_h(k) \ge \frac{1}{2} - 2h\frac{z(\pi)}{2^H}. \tag{19}$$

For the sake of contradiction, we assume $z(\pi) = o\left(\exp(cH)/H\right)$, then for large enough $H$ we have,

$$1/2 - \frac{20Hz(\pi)}{2^H} \ge 0.49,$$

which means that $\sum_{k \in \mathcal{K}} 2^{-H} \cdot \nu_h(k) \ge 0.49$ for all $h \le 10H$. Consequently, for $H > 500$, we have

$$\eta(\pi^\star) - \eta(\pi) \ge \sum_{h=1}^{10H} (0.183 \times 0.49)\gamma^{h-1} \ge 0.089 \cdot \frac{1 - \gamma^{10H}}{1 - \gamma} \ge \frac{0.088}{1 - \gamma}.$$

Now, since $\eta(\pi^\star) \le 1/(1 - \gamma)$, we have $\eta(\pi) < 0.92\eta(\pi^\star)$. Therefore for near-optimal policy $\pi$, $z(\pi) = \Omega\left(\exp(cH)/H\right)$. $\qquad\square$

### B.3    PROOFS OF LEMMA B.3 AND LEMMA B.4

In this section, we present the proofs of two lemmas used in Section B.1

*Proof of Lemma B.3.* Note that for any $k \in \mathcal{K}$, $s^{(H)} = 1, \forall s \in \ell_k$. Now fix a parameter $k \in \mathcal{K}$. Suppose $\pi(s) = a_i$ for $s \in \ell_k$. Then for any $s$ such that $s^{(H+1)} + i \neq 1$, we have

$$V^\star(s) - Q^\star(s, \pi(s)) \geq \gamma^H - \varepsilon.$$

For $H > 500$, we have $\gamma^H - \varepsilon > 0.366$. Therefore,

$$\int_{s \in \ell_k} (V^\star(s) - Q^\star(s, \pi(s)))\, ds \geq \int_{s \in \ell_k} 0.366 \cdot \mathbb{I}[s^{(H+1)} \neq 1 - i]\, ds \geq 0.366 \cdot 2^{-H-1} = 0.183 \cdot 2^{-H}.$$

$\square$

*Proof of Lemma B.4.* Now let us fix a parameter $H$ and policy $\pi$. For every $h$, we prove by induction that there exists a function $\xi_h(s)$, such that

    (a)  $0 \leq \xi_h(s) \leq \min\{\mu_h^\pi(s), 1\}$,

    (b)  $\inf_{s \in \ell_k} \xi_h(s) = \sup_{s \in \ell_k} \xi_h(s), \quad \forall k \in [A^H]$,

    (c)  $\int_{s \in [0,1)} d\xi_h(s) \geq 1 - h \cdot z(\pi)/2^{H-1}$.

For the base case $h = 1$, we define $\xi_h(s) = \mu_h^\pi(s) = 1$ for all $s \in [0, 1)$. Now we construct $\xi_{h+1}$ from $\xi_h$.

For a fixed $k \in [2^H]$, define $l_k = k \cdot 2^{-H}, r_k = (k + 1) \cdot 2^{-H}$ as the left and right endpoints of interval $\ell_k$. Let $\{x_k^{(i)}\}_{i=1}^2$ be the set of 2 solutions of equation

$$2x + 2^{-H} \equiv l_k \mod 1$$

where $0 \leq x < 1$, and we define $y_k^{(i)} = x_k^{(i)} + 2^{-H} \mod 1$. By definition, only states from the set $\cup_{i=1}^2 [x_k^{(i)}, y_k^{(i)})$ can reach states in interval $\ell_k$ by a single transition. We define a set $I_k = \{i : 1 \leq i \leq 2, |\{\pi(s) : s \in [x_k^{(i)}, y_k^{(i)})\}| = 1\}$. That is, the intervals where policy $\pi$ acts unanimously. Consequently, for $i \in I_k$, the set $\{s : s \in [x_k^{(i)}, y_k^{(i)}), f(s, \pi(s)) \in \ell_k\}$ is an interval of length $2^{-H-1}$, and has the form

$$u_k^{(i)} \stackrel{\text{def}}{=} [x_k^{(i)} + w_k^{(i)} \cdot 2^{-H-1}, x_k^{(i)} + (w_k^{(i)} + 1) \cdot 2^{-H-1})$$

for some integer $w_k^{(i)} \in \{0, 1\}$. By statement (b) of induction hypothesis,

$$\inf_{s \in u_k^{(i)}} \xi_h(s) = \sup_{s \in u_k^{(i)}} \xi_h(s). \tag{20}$$

Now, the density $\xi_{h+1}(s)$ for $s \in \ell_k$ is defined as,

$$\xi_{h+1}(s) \stackrel{\text{def}}{=} \sum_{i \in I_k} \frac{1}{2} \cdot \xi_h(x_k^{(i)} + w_k^{(i)} \cdot 2^{-H-1})$$

The intuition of the construction is that, we discard those density that cause non-uniform behavior (that is, the density in intervals $[x_k^{(i)}, y_k^{(i)})$ where $i \notin I_k$). When the number of pieces of $\pi$ is small, we can keep most of the density. Now, statement (b) is naturally satisfied by definition of $\xi_{h+1}$. We verify statement (a) and (c) below.

For any set $B \subseteq \ell_k$, let $(\mathcal{T}^\pi)^{-1}(B) = \{s \in \mathcal{S} : f(s, \pi(s)) \in B\}$ be the inverse of Markov transition $\mathcal{T}^\pi$. Then we have,

$$(\mathcal{T}^\pi \xi_h)(B) \stackrel{\text{def}}{=} \xi_h\left((\mathcal{T}^\pi)^{-1}(B)\right) = \sum_{i \in \{1,2\}} \xi_h\left((\mathcal{T}^\pi)^{-1}(B) \cap [x_k^{(i)}, y_k^{(i)})\right)$$

$$\geq \sum_{i \in I_k} \xi_h\left((\mathcal{T}^\pi)^{-1}(B) \cap [x_k^{(i)}, y_k^{(i)})\right)$$

$$= \sum_{i \in I_k} \left|(\mathcal{T}^\pi)^{-1}(B) \cap [x_k^{(i)}, y_k^{(i)})\right| \xi_h\left(x_k^{(i)} + w_k^{(i)} \cdot 2^{-H-1}\right) \qquad \text{(By Eq. (20))}$$

$$= \sum_{i \in I_k} \frac{|B|}{2} \xi_h\left(x_k^{(i)} + w_k^{(i)} \cdot 2^{-H-1}\right),$$

where $|\cdot|$ is the shorthand for standard Lebesgue measure.

By definition, we have

$$\xi_{h+1}(B) = \sum_{i \in I_k} \frac{|B|}{2} \xi_h\left(x_k^{(i)} + w_k^{(i)} \cdot 2^{-H-1}\right) \leq (\mathcal{T}^\pi \xi_h)(B) \leq (\mathcal{T}^\pi \mu_h^\pi)(B) = \mu_{h+1}^\pi(B),$$

which verifies statement (a).

For statement (c), recall that $\mathcal{S} = [0, 1)$ is the state space. Note that $\mathcal{T}^\pi$ preserve the overall density. That is $(\mathcal{T}^\pi \xi_h)(\mathcal{S}) = \xi_h(\mathcal{S})$. We only need to prove that

$$(\mathcal{T}^\pi \xi_h)(\mathcal{S}) - \xi_{h+1}(\mathcal{S}) \leq h \cdot z(\pi)/2^{H-1} \qquad (21)$$

and statement (c) follows by induction.

By definition of $\xi_{h+1}(s)$ and the induction hypothesis that $\xi_h(s) \leq 1$, we have

$$(\mathcal{T}^\pi \xi_h)(\ell_k) - \xi_{h+1}(\ell_k) \leq (2 - |I_k|)2^{-H}.$$

On the other hand, for any $s \in \mathcal{S}$, the set $\{k \in [2^H] : s \in \cup_{i=1}^2 [x_k^{(i)}, y_k^{(i)})\}$ has cardinality 2, which means that one intermittent point of $\pi$ can correspond to at most 2 intervals that are not in $I_k$ for some $k$. Thus, we have

$$\sum_{0 \leq k < 2^H} |I_k| \geq 2^{H+1} - \sum_{s: \pi^-(s) \neq \pi^+(s)} \left|\{k \in [2^H] : s \in \cup_{i=1}^2 [x_k^{(i)}, y_k^{(i)})\}\right| \geq 2^{H+1} - 2 \cdot z(\pi).$$

Consequently

$$(\mathcal{T}^\pi \xi_h)(\mathcal{S}) - \xi_{h+1}(\mathcal{S}) = \sum_{0 \leq k < 2^H} ((\mathcal{T}^\pi \xi_h)(\ell_k) - \xi_{h+1}(\ell_k)) \leq z(\pi)2^{-H+1},$$

which proves statement (c). $\qquad\qquad\square$

### B.4 Sample Complexity Lower Bound of Q-learning

Recall that corollary 4.4 says that in order to find a near-optimal policy by a Q-learning algorithm, an exponentially large Q-network is required. In this subsection, we show that even if an exponentially large Q-network is applied for $Q$ learning, still we need to collect an exponentially large number of samples, ruling out the possibility of efficiently solving the constructed MDPs with Q-learning algorithms.

Towards proving the sample complexity lower bound, we consider a stronger family of Q-learning algorithm, *Q-learning with Oracle* (Algorithm 3). We assume that the algorithm has access to a Q-ORACLE, which returns the optimal $Q$-function upon querying any pair $(s, a)$ during the training process. *Q-learning with Oracle* is conceptually a stronger computation model than the vanilla Q-learning algorithm, because it can directly fit the $Q$ functions with supervised learning, without relying on the rollouts or the previous $Q$ function to estimate the target $Q$ value. Theorem B.5 proves a sample complexity lower bound for Q-learning algorithm on the constructed example.

---

**Algorithm 3** Q-LEARNING WITH ORACLE

---

**Require:** A hypothesis space $\mathcal{Q}$ of $Q$-function parameterization.
 1: Sample $s_0 \sim \mu$ from the initial state distribution $\mu$
 2: **for** $i = 1, 2, \cdots, n$ **do**
 3:     Decide whether to restart the trajectory by setting $s_i \sim \mu$ based on historical information
 4:     Query Q-ORACLE to get the function $Q^\star(s_i, \cdot)$.
 5:     Apply any action $a_i$ (according to any rule) and sample $s_{i+1} \sim f(s_i, a_i)$.
 6: Learn the $Q$-function that fit all the data the best:
$$Q \leftarrow \arg\min_{Q \in \mathcal{Q}} \frac{1}{n} \sum_{i=1}^{n} \left(Q(s_i, a_i) - Q^\star(s_i, a_i)\right)^2 + \lambda R(Q)$$
 7: Return the greedy policy according to $Q$.

---

**Theorem B.5** (Informal Version of Theorem B.7). *Suppose $\mathcal{Q}$ is an infinitely-wide two-layer neural networks, and $R(Q)$ is $\ell_1$ norm of the parameters and serves as a tiebreaker. Then, any instantiation of the* Q-LEARNING WITH ORACLE *algorithm requires exponentially many samples to find a policy $\pi$ such that $\eta(\pi) > 0.99\eta(\pi^\star)$.*

Formal proof of Theorem B.5 is given in Appendix B.5. The proof of Theorem B.5 is to exploit the sparsity of the solution found by minimal-norm tie-breaker. It can be proven that there are at most $O(n)$ non-zero neurons in the minimal-norm solution, where $n$ is the number of data points. The proof is completed by combining with Theorem 4.3.

## B.5    PROOF OF THEOREM B.5

A two-layer ReLU neural net $Q(s, \cdot)$ with input $s$ is of the following form,

$$Q(s, a) = \sum_{i=1}^{d} w_{i,a} \left[k_i s + b_i\right]_+ + c_a, \tag{22}$$

where $d$ is the number of hidden neurons. $w_{i,a}, c_a, k_i, b_i$ are parameters of this neural net, where $c_{i,a}, b_i$ are bias terms. $[x]_+$ is a shorthand for ReLU activation $\mathbb{I}[x > 0]x$. Now we define the norm of a neural net.

**Definition B.6** (Norm of a Neural Net). *The norm of a two-layer ReLU neural net is defined as,*

$$\sum_{i=1}^{d} |w_{i,a}| + |k_i|. \tag{23}$$

Recall that the *Q-learning with oracle* algorithm finds the solution by the following supervised learning problem,

$$\min_{Q \in \mathcal{Q}} \frac{1}{n} \sum_{t=1}^{n} \left(Q(s_t, a_t) - Q^\star(s_t, a_t)\right)^2. \tag{24}$$

Then, we present the formal version of theorem B.5.

**Theorem B.7.** *Let $Q$ be the minimal $\ell_1$ norm solution to Eq. (24), and $\pi$ the greedy policy according to Q. When $n = o(\exp(cH)/H)$, we have $\eta(\pi) < 0.99\eta(\pi^\star)$.*

The proof of Theorem B.5 is by characterizing the minimal-norm solution, namely the sparsity of the minimal-norm solution as stated in the next lemma.

**Lemma B.8.** *The minimal-norm solution to Eq. (24) has at most $32n + 1$ non-zero neurons. That is, $|\{i : k_i \neq 0\}| \leq 32n + 1$.*

We first present the proof of Theorem B.7, followed by the proof of Theorem B.8.

*Proof of Theorem B.7.* Recall that the policy is given by $\pi(s) = \arg\max_{a \in \mathcal{A}} Q(s, a)$. For a $Q$-function with $32n + 2$ pieces, the greedy policy according to $Q(s, a)$ has at most $64n + 4$ pieces. Combining with Theorem 4.3, in order to find a policy $\pi$ such that $\eta(\pi) > 0.99\eta(\pi^\star)$, $n$ needs to be exponentially large (in effective horizon $H$). $\qquad\square$

Proof of Lemma B.8 is based on merging neurons. Let $x_i = -b_i/k_i, \mathbf{w}_i = (w_{i,1}, w_{i,2})$, and $\mathbf{c} = (c_1, c_2)$. In vector form, neural net defined in Eq. (22) can be written as,

$$Q(s, \cdot) = \sum_{i=1}^{d} \mathbf{w}_i \left[ k_i (s - x_i) \right]_+ + \mathbf{c}.$$

First we show that neurons with the same $x_i$ can be merged together.

**Lemma B.9.** *Consider the following two neurons,*

$$k_1 \left[ s - x_1 \right]_+ \mathbf{w}_1, \quad k_2 \left[ s - x_2 \right]_+ \mathbf{w}_2.$$

*with $k_1 > 0$, $k_2 > 0$. If $x_1 = x_2$, then we can replace them with one single neuron of the form $k' \left[ x - x_1 \right]_+ \mathbf{w}'$ without changing the output of the network. Furthermore, if $\mathbf{w}_1 \neq 0, \mathbf{w}_2 \neq 0$, the norm strictly decreases after replacement.*

*Proof.* We set $k' = \sqrt{|k_1 \mathbf{w}_1 + k_2 \mathbf{w}_2|_1}$, and $w' = (k_1 \mathbf{w}_1 + k_2 \mathbf{w}_2)/k'$, where $|\mathbf{w}|_1$ represents the 1-norm of vector $\mathbf{w}$. Then, for all $s \in \mathbb{R}$,

$$k' \left[ x - x_1 \right]_+ \mathbf{w}' = (k_1 \mathbf{w}_1 + k_2 \mathbf{w}_2) \left[ s - x_1 \right]_+ = k_1 \left[ s - x_1 \right]_+ \mathbf{w}_1 + k_2 \left[ s - x_1 \right]_+ \mathbf{w}_2.$$

The norm of the new neuron is $|k'| + |\mathbf{w}'|_1$. By calculation we have,

$$|k'| + |\mathbf{w}'|_1 = 2\sqrt{|k_1 \mathbf{w}_1 + k_2 \mathbf{w}_2|_1} \leq 2\sqrt{|k_1 \mathbf{w}_1|_1 + |k_2 \mathbf{w}_2|_1}$$

$$\overset{(a)}{\leq} 2\left( \sqrt{|k_1 \mathbf{w}_1|_1} + \sqrt{|k_2 \mathbf{w}_2|_1} \right) \leq |k_1| + |\mathbf{w}_1|_1 + |k_2| + |\mathbf{w}_2|_1.$$

Note that the inequality (a) is strictly less when $|k_1 \mathbf{w}_1|_1 \neq 0$ and $|k_2 \mathbf{w}_2|_1 \neq 0$. $\qquad\square$

Next we consider merging two neurons with different intercepts between two data points. Without loss of generality, assume the data points are listed in ascending order. That is, $s_i \leq s_{i+1}$.

**Lemma B.10.** *Consider two neurons*

$$k_1 \left[ s - x_0 \right]_+ \mathbf{w}_1, \quad k_2 \left[ s - x_0 - \delta \right]_+ \mathbf{w}_2.$$

*with $k_1 > 0, k_2 > 0$. If $s_i \leq x_0 < x_0 + \delta \leq s_{i+1}$ for some $1 \leq i \leq n$, then the two neurons can replaced by a set of three neurons,*

$$k' \left[ s - x_0 \right]_+ \mathbf{w}', \quad \tilde{k} \left[ s - s_i \right]_+ \tilde{\mathbf{w}}, \quad \tilde{k} \left[ s - s_{i+1} \right]_+ (-\tilde{\mathbf{w}})$$

*such that for $s \leq s_i$ or $s \geq s_{i+1}$, the output of the network is unchanged. Furthermore, if $\delta \leq (s_{i+1} - s_i)/16$ and $|\mathbf{w}_1|_1 \neq 0, |\mathbf{w}_2|_1 \neq 0$, the norm decreases strictly.*

*Proof.* For simplicity, define $\Delta = s_{i+1} - s_i$. We set

$$k' = \sqrt{|k_1 \mathbf{w}_1 + k_2 \mathbf{w}_2|_1},$$
$$\mathbf{w}' = (k_1 \mathbf{w}_1 + k_2 \mathbf{w}_2)/k',$$
$$\tilde{k} = \sqrt{|k_2 \mathbf{w}_2|_1 \delta/\Delta},$$
$$\tilde{\mathbf{w}} = -k_2 \mathbf{w}_2 \delta/(\Delta \tilde{k}).$$

Note that for $s \leq s_i$, all of the neurons are inactive. For $s \geq s_{i+1}$, all of the neurons are active, and

$$k' \mathbf{w}'(s - x_0) + \tilde{k} \tilde{\mathbf{w}}(s - s_i) - \tilde{k} \tilde{\mathbf{w}}(s - s_{i+1})$$
$$= (k_1 \mathbf{w}_1 + k_2 \mathbf{w}_2)(s - x_0) - k_2 \mathbf{w}_2 \delta$$
$$= k_1(s - x_0)\mathbf{w}_1 + k_2(s - x_0 - \delta)\mathbf{w}_2,$$

which means that the output of the network is unchanged. Now consider the norm of the two networks. Without loss of generality, assume $|k_1 \mathbf{w}_1|_1 > |k_2 \mathbf{w}_2|_1$. The original network has norm $|k_1| + |\mathbf{w}_1|_1 + |k_2| + |\mathbf{w}_2|_1$. And the new network has norm

$$|k'| + |\mathbf{w}'|_1 + 2|\tilde{k}| + 2|\tilde{\mathbf{w}}|_1 = 2\sqrt{|k_1 \mathbf{w}_1 + k_2 \mathbf{w}_2|_1} + 4\sqrt{|k_2 \mathbf{w}_2|_1 \delta/\Delta}$$

$$\overset{(a)}{\leq} |k_1| + |\mathbf{w}_1|_1 + |k_2| + |\mathbf{w}_2|_1 + \left( 4\sqrt{|k_2 \mathbf{w}_2|_1 \delta/\Delta} - \frac{1}{2}(|k_2| + |\mathbf{w}_2|_1) \right),$$

where the inequality (a) is a result of Lemma E.1, and is strictly less when $|\mathbf{w}_1|_1 \neq 0, |\mathbf{w}_2|_1 \neq 0$.

When $\delta/\Delta < 1/16$, we have $\left(4\sqrt{|k_2\mathbf{w}_2|_1\delta/\Delta} - \frac{1}{2}(|k_2| + |\mathbf{w}_2|_1)\right) < 0$, which implies that

$$|k'| + |\mathbf{w}'|_1 + 2|\tilde{k}| + 2|\tilde{\mathbf{w}}|_1 < |k_1| + |\mathbf{w}_1|_1 + |k_2| + |\mathbf{w}_2|_1.$$

$\square$

Similarly, two neurons with $k_1 < 0$ and $k_2 < 0$ can be merged together.

Now we are ready to prove Lemma B.8. As hinted by previous lemmas, we show that between two data points, there are at most 34 non-zero neurons in the minimal norm solution.

*Proof of Lemma B.8.* Consider the solution to Eq. (24). Without loss of generality, assume that $s_i \leq s_{i+1}$. In the minimal norm solution, it is obvious that $|\mathbf{w}_i|_1 = 0$ if and only if $k_i = 0$. Therefore we only consider those neurons with $k_i \neq 0$, denoted by index $1 \leq i \leq d'$.

Let $\mathcal{B}_t = \{-b_i/k_i : 1 \leq i \leq d', s_t < -b_i/k_i < s_{t+1}, k_i > 0\}$. Next we prove that in the minimal norm solution, $|\mathcal{B}_t| \leq 15$. For the sake of contradiction, suppse $|\mathcal{B}_t| > 15$. Then there exists $i, j$ such that, $s_t < -b_i/k_i < s_{t+1}, s_t < -b_j/k_j < s_{t+1}, |b_i/k_i - b_j/k_j| < (s_{t+1} - s_i)/16$, and $k_i > 0, k_j > 0$. By Lemma B.10, we can obtain a neural net with smaller norm by merging neurons $i, j$ together without violating Eq. (24), which leads to contradiction.

By Lemma B.9, $|\mathcal{B}_t| \leq 15$ implies that there are at most 15 non-zero neurons with $s_t < -b_i/k_i < s_{t+1}$ and $k_i > 0$. For the same reason, there are at most 15 non-zero neurons with $s_t < -b_i/k_i < s_{t+1}$ and $k_i < 0$.

On the other hand, there are at most 2 non-zero neurons with $s_t = -b_i/k_i$ for all $t \leq n$, and there are at most 1 non-zero neurons with $-b_i/k_i < s_1$. Therefore, we have $d' \leq 32n + 1$. $\square$

## B.6 PROOF OF THEOREM 5.1

In this section we present the full proof of Theorem 5.1.

*Proof.* First we define the true trajectory estimator

$$\eta(s_0, a_0, a_1, \cdots, a_k) = \sum_{j=0}^{k-1} \gamma^j r(s_j, a_j) + \gamma^k Q^\star(s_k, a_k),$$

the true optimal action sequence

$$a_0^\star, a_1^\star, \cdots, a_k^\star = \arg\max_{a_0, a_1, \cdots, a_k} \eta(s_0, a_0, a_1, \cdots, a_k),$$

and the true optimal trajectory

$$s_0^\star = s_0, \ s_j^\star = f(s_{j-1}^\star, a_{j-1}^\star), \forall j > 1.$$

It follows from the definition of optimal policy that, $a_j^\star = \pi^\star(s_j)$. Consequently we have

$$s_k^{(H-k+1)} = s_k^{(H-k+2)} = \cdots = s_k^{(H)} = 1.$$

Define the set $G = \{s : s^{(H-k+1)} = s^{(H-k+2)} = \cdots = s^{(H)} = 1\}$. We claim that the following function satisfies the statement of Theorem 5.1

$$Q(s, a) = \mathbb{I}[s \in G] \cdot \frac{2}{1 - \gamma}.$$

Since $s_k^\star \in G$, and $s_k \notin G$ for $s_k$ generated by non-optimal action sequence, we have

$$Q(s_k^\star, a) > Q^\star(s_k^\star, a) \geq Q^\star(s_k, a) > Q(s_k, a),$$

where the second inequality comes from the optimality of action sequence $a_h^\star$. As a consequence, for any $(a_0, a_1, \cdots, a_k) \neq (a_0^\star, a_1^\star, \cdots, a_k^\star)$

$$\hat{\eta}(s_0, a_0^\star, a_1^\star, \cdots, a_k^\star) > \eta(s_0, a_0^\star, a_1^\star, \cdots, a_k^\star) \geq \eta(s_0, a_0, a_1, \cdots, a_k) > \hat{\eta}(s_0, a_0, a_1, \cdots, a_k).$$

Therefore, $(\hat{a}_0^\star, \hat{a}_1^\star, \cdots, \hat{a}_k^\star) = (a_0^\star, a_1^\star, \cdots, a_k^\star)$. $\square$

## C    EXTENSION OF THE CONSTRUCTED FAMILY

In this section, we present an extension to our construction such that the dynamics is Lipschitz. The action space is $\mathcal{A} = \{0, 1, 2, 3, 4\}$. We define $\mathrm{CLIP}(x) = \max\{\min\{x, 1\}, 0\}$.

**Definition C.1.** *Given effective horizon* $H = (1 - \gamma)^{-1}$, *we define an MDP* $M'_H$ *as follows. Let* $\kappa = 2^{-H}$. *The dynamics is defined as*

$$f(s, 0) = \mathrm{CLIP}(2s), \quad f(s, 1) = \mathrm{CLIP}(2s - 1),$$
$$f(s, 2) = \mathrm{CLIP}(2s + \kappa), \quad f(s, 3) = \mathrm{CLIP}(2s + \kappa - 1), \quad f(s, 4) = \mathrm{CLIP}(2s + \kappa - 2).$$

*Reward function is given by*

$$r(s, 0) = r(s, 1) = \mathbb{I}[1/2 \le s < 1]$$
$$r(s, 2) = r(s, 3) = r(s, 4) = \mathbb{I}[1/2 \le s < 1] - 2(\gamma^{H-1} - \gamma^H)$$

The intuition behind the extension is that, we perform the mod operation manually. The following theorem is an analog to Theorem 4.2.

**Theorem C.2.** *The optimal policy* $\pi^\star$ *for* $M'_H$ *is defined by,*

$$\pi^\star(s) = \begin{cases} 0, & \mathbb{I}[s^{(H+1)} = 0] \text{ and } 2s < 1, \\ 1, & \mathbb{I}[s^{(H+1)} = 0] \text{ and } 1 \le 2s < 2, \\ 2, & \mathbb{I}[s^{(H+1)} = 1] \text{ and } 2s + \theta < 1, \\ 3, & \mathbb{I}[s^{(H+1)} = 1] \text{ and } 1 \le 2s + \theta < 2, \\ 4, & \mathbb{I}[s^{(H+1)} = 1] \text{ and } 2 < 2s + \theta. \end{cases} \tag{25}$$

*And the corresponding optimal value function is,*

$$V^\star(s) = \sum_{h=1}^H \gamma^{h-1} s^{(h)} + \sum_{h=H+1}^\infty \gamma^{h-1} \left(1 + 2(s^{(h+1)} - s^{(h)})\right) + \gamma^{H-1} \left(2s^{(H+1)} - 2\right). \tag{26}$$

We can obtain a similar upper bound on the performance of policies with polynomial pieces.

**Theorem C.3.** *Let* $M_H$ *be the MDP constructed in Definition C.1. Suppose a piecewise linear policy* $\pi$ *has a near optimal reward in the sense that* $\eta(\pi) \ge 0.99 \cdot \eta(\pi^\star)$, *then it has to have at least* $\Omega\left(\exp(cH)/H\right)$ *pieces for some universal constant* $c > 0$.

The proof is very similar to that for Theorem 4.3. One of the difference here is to consider the case where $f(s, a) = 0$ or $f(s, a) = 1$ separately. Attentive readers may notice that the dynamics where $f(s, a) = 0$ or $f(s, a) = 1$ may destroy the "near uniform" behavior of state distribution $\mu_h^\pi$ (see Lemma B.4). Here we show that such destroy comes with high cost. Formally speaking, if the clip is triggered in an interval, then the averaged single-step suboptimality gap is $0.1/(1 - \gamma)$.

**Lemma C.4.** *Let* $\ell_k = [k/2^{H/2}, (k+1)/2^{H/2})$. *For* $k \in [2^{H/2}]$, *if policy* $\pi$ *does not change its action at interval* $\ell_k$ *(that is,* $|\{\pi(s) : s \in \ell_k\}| = 1$) *and* $f(s, \pi(s)) = 0$, $\forall s \in \ell_k$ *or* $f(s, \pi(s)) = 1$, $\forall s \in \ell_k$. *We have*

$$\frac{1}{|\ell_k|} \int_{s \in \ell_k} (V^\star(s) - Q^\star(s, \pi(s))) \, ds \ge \frac{0.1}{1 - \gamma} \tag{27}$$

*for large enough* $H$.

*Proof.* Without loss of generality, we consider the case where $f(s, \pi(s)) = 0$. The proof for $f(s, \pi(s)) = 1$ is essentially the same.

By elementary manipulation, we have

$$V^\star(s) - V^\star(0) \ge \sum_{i=1}^H \gamma^{i-1} s^{(i)}.$$

Let $\hat{s} = f(s, \pi^\star(s))$. It follows from Bellman equation (1) that

$$V^\star(s) = r(s, \pi^\star(s)) + \gamma V^\star(\hat{s}),$$
$$Q^\star(s, \pi(s)) = r(s, \pi(s)) + \gamma V^\star(0).$$

Recall that we define $\epsilon = 2\left(\gamma^{H-1} - \gamma^H\right)$. As a consequence,

$$(V^\star(s) - Q^\star(s, \pi(s))) > r(s, \pi^\star(s)) - r(s, \pi(s)) + \gamma(V^\star(\hat{s}) - V^\star(0))$$

$$\geq -\epsilon + \gamma \sum_{i=1}^{H} \gamma^{i-1} \hat{s}^{(i)}.$$

Plugging into Eq (27), we have

$$\frac{1}{|\ell_k|} \int_{s \in \ell_k} (V^\star(s) - Q^\star(s, \pi(s)))\, ds \geq -\epsilon + \frac{1}{|\ell_k|} \int_{s \in \ell_k} \left(\sum_{i=1}^{H} \gamma^i\right) \hat{s}^{(i)}\, ds$$

$$\geq -\epsilon + \sum_{i=1}^{H} \gamma^i \left(\frac{1}{|\ell_k|} \int_{s \in \ell_k} \hat{s}^{(i)}\, ds\right) \geq -\epsilon + \frac{\gamma^{H/2} - \gamma^H}{1 - \gamma}.$$

Lemma 27 is proved by noticing for large enough $H$,

$$-\epsilon + \frac{\gamma^{H/2} - \gamma^H}{1 - \gamma} > \frac{0.1}{1 - \gamma}.$$

$\square$

Let $D = \{0, 1\}$ for simplicity. For any policy $\pi$, we define a transition operator $\hat{\mathcal{T}}^\pi$, such that

$$\left(\hat{\mathcal{T}}^\pi \mu\right)(Z) = \mu\left(\{s : p(s, a) \in Z, f(s, \pi(s)) \notin D\}\right),$$

and the state distribution induced by it, defined recursively by

$$\hat{\mu}_1^\pi(s) = 1,$$
$$\hat{\mu}_h^\pi = \hat{\mathcal{T}}^\pi \mu_{h-1}^\pi.$$

We also define the density function for states that are truncated as follows,

$$\hat{\rho}_h^\pi(s) = \mathbb{I}[f(s, \pi(s)) \in D]\hat{\mu}_h^\pi(s).$$

Following advantage decomposition lemma (Corollary B.2), the key step for proving Theorem C.3 is

$$\eta(\pi^\star) - \eta(\pi) \geq \sum_{h=1}^{\infty} \gamma^{h-1} \mathbb{E}_{s \sim \hat{\mu}_h^\pi} [V^\star(s) - Q^\star(s, \pi(s))] + \sum_{h=1}^{\infty} \gamma^h \mathbb{E}_{s \sim \rho_h^\pi} [V^\star(s) - Q^\star(s, \pi(s))].$$

(28)

Similar to Lemma B.4, the following lemma shows that the density for most of the small intervals is either uniformly clipped, or uniformly spread over this interval.

**Lemma C.5.** *Let $z(\pi)$ be the number of pieces of policy $\pi$. For $k \in [2^{H/2}]$, define interval $\ell_k = [k/2^{H/2}, (k+1)/2^{H/2})$. Let $\nu_h(k) = \inf_{s \in \ell_k} \hat{\mu}_h^\pi(s)$ and $\omega_h(k) = \inf_{s \in \ell_k} \hat{\rho}_h^\pi(s)$. If the initial state distribution $\mu$ is uniform distribution, then for any $h \geq 1$,*

$$\sum_{k=0}^{2^{H/2}} 2^{-H/2} \cdot \nu_h(k) + \sum_{h'=1}^{h-1} \sum_{k=0}^{2^{H/2}} 2^{-H/2} \cdot \omega_{h'}(k) \geq 1 - 2h \frac{z(\pi) + 10}{2^{H/2}}.$$

(29)

*Proof.* Omitted. The proof is similar to Lemma B.4. $\square$

Now we present the proof for Theorem C.3.

*Proof of Theorem C.3.* For any $k \in [2^{H/2}]$, consider the interval $\ell_k = [k/2^{H/2}, (k+1)/2^{H/2}) .$. If $\pi$ does not change at interval $\ell_k$ (that is, $|\{\pi(s) : s \in \ell_k\}| = 1$), by Lemma B.3 we have

$$\int_{s \in \ell_k} (V^\star(s) - Q^\star(s, \pi(s)))\, ds \geq 0.075 \cdot 2^{-H/2}. \tag{30}$$

By Eq (28), Eq (30) and Lemma (27), we have

$$\eta(\pi^\star) - \eta(\pi)$$
$$\geq \sum_{h=1}^{H} \gamma^{h-1} \left( \sum_{k=0}^{2^{H/2}} 0.075 \cdot 2^{-H/2} \cdot \nu_h(k) \right) + \sum_{h=1}^{H} \sum_{k=0}^{2^{H/2}} \gamma^h \cdot 2^{-H/2} \cdot \omega_h(k) \cdot \frac{0.1}{1 - \gamma}. \tag{31}$$

By Lemma C.5, we get

$$\sum_{k=0}^{2^{H/2}} 2^{-H/2} \cdot \nu_h(k) + \sum_{h'=1}^{h-1} \sum_{k=0}^{2^{H/2}} 2^{-H/2} \cdot \omega_{h'}(k) \geq 1 - 2h \frac{z(\pi) + 10}{2^{H/2}}. \tag{32}$$

For the sake of contradiction, we assume $z(\pi) = o\left(\exp(cH)/H\right)$, then for large enough $H$ we have,

$$1 - 2 \frac{Hz(\pi) + 10}{2^{H/2}} > 0.8.$$

Consequently,

$$\sum_{k=0}^{2^{H/2}} 2^{-H/2} \cdot \nu_h(k) > 0.8 - \sum_{h'=1}^{h-1} \sum_{k=0}^{2^{H/2}} 2^{-H/2} \cdot \omega_{h'}(k). \tag{33}$$

Plugging in Eq (31), we get

$$\eta(\pi^\star) - \eta(\pi)$$
$$\geq \sum_{h=1}^{H} 0.075\gamma^{h-1} \left( \sum_{k=0}^{2^{H/2}} 2^{-H/2} \nu_h(k) \right) + \sum_{h=1}^{H} \sum_{k=0}^{2^{H/2}} \gamma^h \cdot 2^{-H/2} \cdot \omega_h(k) \cdot \frac{0.1}{1 - \gamma}.$$
$$\geq \sum_{h=1}^{H} 0.075\gamma^{h-1} \left( 0.8 - \sum_{h'=1}^{h-1} \sum_{k=0}^{2^{H/2}} 2^{-H/2} \cdot \omega_{h'}(k) \right) + \sum_{h=1}^{H} \sum_{k=0}^{2^{H/2}} \gamma^h \cdot 2^{-H/2} \cdot \omega_h(k) \cdot \frac{0.1}{1 - \gamma}$$
$$\geq 0.06 \frac{1 - \gamma^H}{1 - \gamma} + \sum_{h=1}^{H} \sum_{k=0}^{2^{H/2}} \cdot 2^{-H/2} \cdot \omega_h(k) \left( \frac{0.1\gamma^h}{1 - \gamma} - 0.075 \sum_{h'=h}^{H} \gamma^{h'-1} \right)$$
$$\geq 0.06 \frac{1 - \gamma^H}{1 - \gamma} + \sum_{h=1}^{H} \sum_{k=0}^{2^{H/2}} \cdot 2^{-H/2} \cdot \omega_h(k) \frac{\gamma^{h-1}}{1 - \gamma} \left( 0.1\gamma - 0.075 \left(1 - \gamma^{H-h}\right) \right)$$

When $\gamma > 1/4$, we have $0.1\gamma - 0.075(1 - \gamma^{H-h}) > 0$. As a consequence,

$$\eta(\pi^\star) - \eta(\pi) > 0.06 \frac{1 - \gamma^H}{1 - \gamma} \geq \frac{0.01}{1 - \gamma}.$$

Now, since $\eta(\pi^\star) \leq 1/(1 - \gamma)$, we have $\eta(\pi) < 0.99\eta(\pi^\star)$. Therefore for near-optimal policy $\pi$, $z(\pi) = \Omega\left(\exp(cH)/H\right).$ $\qquad \square$

## D OMITTED DETAILS OF EMPIRICAL RESULTS IN THE TOY EXAMPLE

### D.1 TWO METHODS TO GENERATE MDPS

In this section we present two methods of generating MDPs. In both methods, the dynamics $p(s, a)$ has three pieces and is Lipschitz. The dynamics is generated by connecting kinks by linear lines.

**RAND method.** As stated in Section 4.3, the RAND method generates kinks $\{x_i\}$ and the corresponding values $\{x'_i\}$ randomly. In this method, the generated MDPs are with less structure. The details are shown as follows.

- State space $\mathcal{S} = [0, 1)$.
- Action space $\mathcal{A} = \{0, 1\}$.
- Number of pieces is fixed to 3. The positions of the kinks are generated by, $x_i \sim U(0, 1)$ for $i = 1, 2$ and $x_0 = 0, x_1 = 1$. The values are generated by $x'_i \sim U(0, 1)$.
- The reward function is given by $r(s, a) = s, \ \forall s \in \mathcal{S}, a \in \mathcal{A}$.
- The horizon is fixed as $H = 10$.
- Initial state distribution is $U(0, 1)$.

Figure 1 visualizes one of the RAND-generated MDPs with complex Q-functions.

**SEMI-RAND method.** In this method, we add some structures to the dynamics, resulting in a more significant probability that the optimal policy is complex. We generate dynamics with fix and shared kinks, generate the output at the kinks to make the functions fluctuating. The details are shown as follows.

- State space $\mathcal{S} = [0, 1)$.
- Action space $\mathcal{A} = \{0, 1\}$.
- Number of pieces is fixed to 3. The positions of the kinks are generated by, $x_i = i/3, \ \forall 0 \leq i \leq 3$. And the values are generated by $x'_i \sim 0.65 \times \mathbb{I}[i \mod 2 = 0] + 0.35 \times U(0, 1)$.
- The reward function is $r(s, a) = s$ for all $a \in \mathcal{A}$.
- The horizon is fixed as $H = 10$.
- Initial state distribution is $U(0, 1)$.

Figure 1 visualizes one of the MDPs generated by SEMI-RAND method.

## D.2 THE COMPLEXITY OF OPTIMAL POLICIES IN RANDOMLY GENERATED MDPS

We randomly generate $10^3$ 1-dimensional MDPs whose dynamics has constant number of pieces. The histogram of number of pieces in optimal policy $\pi^\star$ is plotted. As shown in Figure 9, even for horizon $H = 10$, the optimal policy tends to have much more pieces than the dynamics.

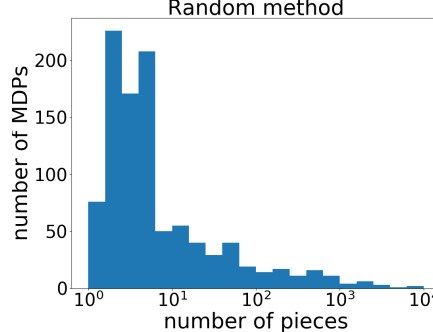
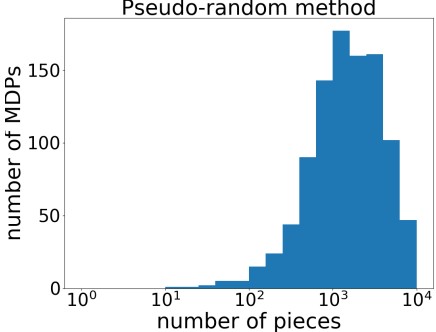

Figure 9: The histogram of number of pieces in optimal policy $\pi^\star$ in random method (left) and semi-random method(right).

## D.3 IMPLEMENTATION DETAILS OF ALGORITHMS IN RANDOMLY GENERATED MDP

**SEMI-RAND MDP** The MDP where we run the experiment is given by the SEMI-RAND method, described in Section D.1. We list the dynamics of this MDP in the following.

$$r(s, a) = s, \quad \forall s \in \mathcal{S}, a \in \mathcal{A},$$

$$f(s,0) = \begin{cases} (0.131 - 0.690) \cdot x/0.333 + 0.690, & 0 \le x < 0.333, \\ (0.907 - 0.131) \cdot (x - 0.333)/0.334 + 0.131, & 0.333 \le x < 0.667, \\ (0.079 - 0.907) \cdot (x - 0.667)/0.333 + 0.907, & 0.667 \le x, \end{cases}$$

$$f(s,1) = \begin{cases} (0.134 - 0.865) \cdot x/0.333 + 0.865, & 0 \le x < 0.333, \\ (0.750 - 0.134) \cdot (x - 0.333)/0.334 + 0.134, & 0.333 \le x < 0.667, \\ (0.053 - 0.750) \cdot (x - 0.667)/0.333 + 0.750, & 0.667 \le x, \end{cases}$$

**Implementation details of DQN algorithm**  We present the hyper-parameters of DQN algorithm. Our implementation is based on PyTorch tutorials[7].

- The Q-network is a fully connected neural net with one hidden-layer. The width of the hidden-layer is varying.
- The optimizer is SGD with learning rate $0.001$ and momentum $0.9$.
- The size of replay buffer is $10^4$.
- Target-net update frequency is $50$.
- Batch size in policy optimization is $128$.
- The behavior policy is greedy policy according to the current Q-network with $\epsilon$-greedy. $\epsilon$ exponentially decays from $0.9$ to $0.01$. Specifically, $\epsilon = 0.01 + 0.89 \exp(-t/200)$ at the $t$-th episode.

**Implementation details of MBPO algorithm**  For the model-learning step, we use $\ell_2$ loss to train our model, and we use Soft Actor-Critic (SAC) (Haarnoja et al., 2018) in the policy optimization step. The parameters are set as,

- number of hidden neurons in model-net: $32$,
- number of hidden neurons in value-net: $512$,
- optimizer for model-learning: Adam with learning rate $0.001$.
- temperature: $\tau = 0.01$,
- the model rollout steps: $M = 5$,
- the length of the rollout: $k = 5$,
- number of policy optimization step: $G = 5$.

Other hyper-parameters are kept the same as DQN algorithm.

**Implementation details of TRPO algorithm**  For the model-learning step, we use $\ell_2$ loss to train our model. Instead of TRPO (Schulman et al., 2015), we use PPO (Schulman et al., 2017) as policy optimizer. The parameters are set as,

- number of hidden neurons in model-net: $32$,
- number of hidden neurons in policy-net: $512$,
- number of hidden neurons in value-net: $512$,
- optimizer: Adam with learning rate $0.001$,
- number of policy optimization step: $5$.
- The behavior policy is $\epsilon$-greedy policy according to the current policy network. $\epsilon$ exponential decays from $0.9$ to $0.01$. Specifically, $\epsilon = 0.01 + 0.89 \exp(-t/20000)$ at the $t$-th episode.

---

[7]https://pytorch.org/tutorials/intermediate/reinforcement_q_learning.html

**Implementation details of Model-based Planning algorithm**    The perfect model-based planning algorithm iterates between learning the dynamics from sampled trajectories, and planning with the learned dynamics (with an exponential time algorithm which enumerates all the possible future sequence of actions). The parameters are set as,

- number of hidden neurons in model-net: 32,
- optimizer for model-learning: Adam with learning rate 0.001.

**Implementation details of bootstrapping**    The training time behavior of the algorithm is exactly like DQN algorithm, except that the number of hidden neurons in the Q-net is set to 64. Other parameters are set as,

- number of hidden neurons in model-net: 32,
- optimizer for model-learning: Adam with learning rate 0.001.
- planning horizon varies.

## E    TECHNICAL LEMMAS

In this section, we present the technical lemmas used in this paper.

**Lemma E.1.** *For $A, B, C, D \geq 0$ and $AC \geq BD$, we have*

$$A + C + \frac{1}{2}(B + D) \geq 2\sqrt{AC + BD}.$$

*Furthermore, when $BD > 0$, the inequality is strict.*

*Proof.*  Note that $A + B + \frac{1}{2}(C + D) \geq 2\sqrt{AC} + \sqrt{BD}$. And we have,

$$\left(2\sqrt{AC} + \sqrt{BD}\right)^2 - \left(2\sqrt{AC + BD}\right)^2 = 4\sqrt{AC \cdot BD} - 3BD \geq BD \geq 0.$$

And when $BD > 0$, the inequality is strict. □

