# OpenReview forum: "Bootstrapping the Expressivity with Model-based Planning"
_ICLR.cc/2020/Conference — Reject_

### Official Review · AnonReviewer3 · 2019-10-21
**Official Blind Review #3**

**Rating:** 6

**Review:**

This paper presents a mainly theoretical argument comparing the expressivity of model-free and model-based RL methods contrary to analysis in the past which usually relies on sample complexity. They construct a family of MDPs, where the true dynamics belong to a simple function class (in terms of the number of linear pieces needed to define the function), but the corresponding optimal Q-function belongs to a function class not necessarily expressible by a simple function. The paper then builds a similar case for randomly/semi-randomly generated MDPs. Finally, they propose to bootstrap the Q-function with n-step returns to boost the expressivity exponentially.

I would lean towards accepting this paper, as this paper looks at an interesting problem and their analysis seems to be rigorous and valuable for the community to build upon. My questions/comments are as follows:

1. Previous work, for example [1], also talks about bootstrapping a neural net Q-function using an n-step approximation so as to improve the efficiency of a pure model-free algorithm. I think this paper should be cited.

2. In terms of the experiments, it is hard to understand the significance and connection to the theory. The theory talks about the expressive power of Q-functions, which suggests that we should look at only the asymptotic performance on these tasks, but most of the results are similar to MBPO or SAC in terms of asymptotic performance, although with a different learning speed, which could have to do with different factors.

3. This paper shows the existence and provides a constructive proof for a family of MDPs where expressing optimal Q-functions is exponentially harder than expressing dynamics. But how likely is such a setting to arise in MDPs in practice? For example, on the gym benchmarks, we would expect fairly not so complicated Q-functions -- although they might take longer to learn.

4. There is also a divide between learning Q*, and learning a policy that optimizes Q* reasonably well. Further, the requirement for the Q-function is only to get relative ordering between Q-values for different actions at states visited by the optimal policy right, which might not be the same as the Q-function landscape across state-action pairs. So, I am not sure if the expressivity of the optimal Q-function, in general, is the right metric to answer this question, but I also completely agree that this is a good starting point.

5. I think when applying RL to the current benchmarks or problems we have, the main problem could be linked to optimization of a neural-net Q-function via bootstrapping (in the sense of approximate dynamic programming) as compared to the expressive power of optimal Q-functions. But I agree that the problem being looked at in the paper would also exist.

References:
[1] Plan Online, Learn Offline: Efficient Learning and Exploration via Model-Based Control, Lowrey et.al.



**Experience Assessment:**

I have read many papers in this area.

**Review Assessment: Checking Correctness Of Derivations And Theory:**

I assessed the sensibility of the derivations and theory.

**Review Assessment: Checking Correctness Of Experiments:**

I assessed the sensibility of the experiments.

**Review Assessment: Thoroughness In Paper Reading:**

I read the paper at least twice and used my best judgement in assessing the paper.

---

> ### Author Response · Authors · 2019-11-14
> **Response to R3**
>
>
> We thank reviewer 3 for the review. The review expressed "valuable rigorous contributions for the community to build upon." The review also provides many constructive suggestions and questions that we will address below. We have updated the paper with the citations that the reviewer suggested and with experiments that are relevant to the reviewer's questions.
>
>
> --- 2 & 3: the connection of the experiment to the theory: "The theory talks about the expressive power of Q-functions, but most of the results are similar to MBPO or SAC in terms of asymptotic performance …", "on the gym benchmarks, we would expect fairly not so complicated Q-functions"
>
> Thanks for the valuable comments. We do believe there is a gap between the asymptotic performances of BOOTS + X vs. X, for X = MBSAC, SAC, or MBPO. We updated Figure 5 with a new result on BOOTS+MBPO. Here we can see that both BOOTS+MBPO and MBPO achieve near asymptotic performance at 400K steps, and there is a gap between them (55k for BOOTS+MBPO vs 5K for MBPO).
>
> (We also like to clarify that, e.g., in Figure 4 (right), BOOTS+SAC (or BOOTS+MBSAC) is constantly much better than SAC (or MBSAC, respectively) throughout the first 400K steps. However, the result does not prove or refute that BOOTS+SAC and SAC have similar asymptotic performances, because the SAC and MBSAC have not fully converged yet. We are running longer experiments and will update in one or two days.)
>
> --- "4. … So, I am not sure if the expressivity of the optimal Q-function, in general, is the right metric to answer this question, but I also completely agree that this is a good starting point. "
>
> We agree with the reviewer's point that the expressivity of optimal Q-function is not necessarily the right metric. However, we did also characterize the expressivity of the policy network in Theorem 4.3: the first part of Theorem 4.3 says that any policy network with good reward needs to have exponential size. Concretely, we proved that: a) any policy network with a near-optimal reward requires exponential pieces, and b) so does any Q-function which can induce a near-optimal policy. So here in (b), we do not require the Q-function to be correct at every state-action pair. As long as the Q-function is capable of inducing a good policy, then it needs to have an exponential number of pieces.)

---

### Official Review · AnonReviewer2 · 2019-10-23
**Official Blind Review #2**

**Rating:** 3

**Review:**

The paper highlights an interesting issue regarding approximability of function approximators (neural networks). The paper provides cases where the action value function is difficult to approximate and is much more difficult than the dynamics of a model. The author conducts some experiments to claim that even with a large NN, DQN still finds a suboptimal policy. Theorems regarding the appoximability of the action value function are presented. Then the paper proposes that rollout-based search should be preferred for planning and conducts some experiments to verify this. Although the paper points out interesting issues of approximating action-value function, both the motivation and the suggestion regarding MBRL are not convincing.

1. In term of the motivation, those cases listed in the paper are interesting, but they are not representative. In fact, the Dynamics can be far more complicated and it is still an open problem regarding how to learn the Dynamics. Furthermore, the proposed method is to simply combine MCTS and bootstrap value estimates. The method itself is not novel and it basically down weights the bootstrap estimate. It is very intuitive that some appropriate combination between the two can yield better performance. However, in model-based setting, the reward sequence can be highly variant and non-stationary, there is no solid reason to believe this can be always better. The motivating experiments in figure 3 are not persuasive. There can be many reasons for a deep RL algorithm to find a suboptimal policy: boostrap target interference, overestimation, difficulty of optimization, etc. It is quite confusing which factor leads to suboptimal performance of DQN.

2. Theorem 4.3 does not make sense to me. What does it mean by “no constant depth NN can approximate the optimal policy with near optimal rewards?” Notice that, in machine learning community, people rarely pursue perfect approximation (equality). As long as the approximation error can be reasonably small, the approximator should be still useful. Does “no constant depth NN can approximate the optimal policy” mean the approximation error is unbounded? I believe we can still expect a large NN to approximate the action-value function very well.

A minor issue. In page 2, the paper writes “model-free RL or MB policy optimization suffer from …, whereas MB planning … ”. MB planning is not a separate category of MB policy optimization. Such statement is not accurate.


**Experience Assessment:**

I have read many papers in this area.

**Review Assessment: Checking Correctness Of Derivations And Theory:**

I assessed the sensibility of the derivations and theory.

**Review Assessment: Checking Correctness Of Experiments:**

I assessed the sensibility of the experiments.

**Review Assessment: Thoroughness In Paper Reading:**

I read the paper at least twice and used my best judgement in assessing the paper.

---

> ### Author Response · Authors · 2019-11-14
> **Response to R2**
>
> We thank the anonymous reviewer 2 for the review! We address the comments of the reviewer below.
>
> --- “In fact, the Dynamics can be far more complicated and it is still an open problem regarding how to learn the Dynamics. “
>
> We respectfully disagree with that the dynamics should always be considered as complicated and hard to learn: our experiments (especially the ones on Humanoid) actually shows, contrary to the conventional wisdom, that the dynamics can be less complicated than the Q-function in the following sense. e.g., Figure 4 (right) shows that even for the Humanoid environment, whose dynamics was considered as very complicated conventionally, planning with a learned dynamics gives a huge boost even in the early stage of the training (e.g., at 100K step, the improvement is from 1K reward to 3.5K reward). This suggests at 100K step, the dynamics is already learned well (at least well enough to be significantly useful in planning. ) However, the Q-function at 100K steps is only able to give about 1K reward. .
>
> --- “The motivating experiments in figure 3 are not persuasive. There can be many reasons for a deep RL algorithm to find a suboptimal policy: boostrap target interference, overestimation, difficulty of optimization, etc. It is quite confusing which factor leads to suboptimal performance of DQN.”
>
> We first clarify that the experiment in Figure 3 is NOT supposed to rigorously prove that expressivity is the culprit. In our case, (and as in many other cases), empirical results are supposed to verify the plausibility of an explanation/conjecture, but inevitably cannot rule out all other possible explanations. In particular, for expressivity, we are not aware of any empirical approach that can rigorously rule out the existence of a good neural network approximation. The only substantial evidence would be a mathematical proof, which is what we did in the theoretical part of the paper. Thus, in our opinion, the experiments here are only to complement the theory to show that empirical observation is consistent with theoretical predictions.
>
> Therefore, our experiment in Figure 3 is supposed to complement the main theorems, which in turn proves that the expressivity is a bottleneck. Our Figure 3 (left) is consistent with the Theorem 4.3 that proves the expressivity limitations, and our Figure 3 (right) is consistent with the Theorem 5.1 that shows the planning can help improve the expressivity.
>
> --- “Theorem 4.3 does not make sense to me. What does it mean by “no constant depth NN [with polynomial width] can approximate the optimal policy with near optimal rewards? Notice that, in machine learning community, people rarely pursue perfect approximation (equality) …. ”
>
> The phrase “near optimal rewards” means a constant approximation of the optimal reward as the same as the previous mention in the same theorem statement. We would also like to kindly point out that the reviewer missed the phrase “with polynomial width” in the quote of our statement, which might cause confusion. Concretely, we meant that no neural networks with polynomial width in H can represent a policy $\pi$ with the property that $\eta(\pi) \ge 0.99 \eta(\pi^*)$. In other words, the approximation ratio (the ratio between the reward of the policy and the optimal reward) cannot approach 1, as we have a larger and larger network, as long as the network size remains a polynomial in H.
>
> We also note that we didn’t spend effort on optimizing the constant 0.99 here because we considered the main point to be demonstrating that the approximation ratio cannot get arbitrarily close to 1. However, with some simple modification of the analysis, we can improve the ratio from 0.99 to 0.92 (as shown in the revision), and with more effort, we can also improve it to about 0.8.
>
> --- “A minor issue. In page 2, the paper writes “model-free RL or MB policy optimization suffer from …, whereas MB planning … ”. MB planning is not a separate category of MB policy optimization. Such statement is not accurate. “
>
> Here we are making the distinction of MB planning and MB policy optimization based on the (perhaps informal) convention that a) planning refers to algorithms that optimize a sequence of actions online in the test time in a closed-loop fashion; b) policy optimization refers to algorithms in which a policy is obtained in the training in advance, and is applied directly in the test time. According to this definition, we thought that MB planning shouldn’t be a subset of MB policy optimization. We will clarify our use of the terms more carefully and explicitly. The key message of the paper is that planning provides more expressivity than a fixed policy network prepared in advance in training. (EDIT: this para was slightly edited per the discussion below.)

---

> > ### Comment · AnonReviewer2 · 2019-11-15
> > **Thanks for the response.**
> >
> > Thank you for response and I am glad to see it.
> >
> > Let me clarify what my main concern is. In general, the statement the author is trying to establish is indeed quite interesting as I mentioned, and potentially has a large impact. (Given that the author can strengthen the argument.)
> >
> > I am not satisfied with below reasoning line:
> >
> > example of some value function which is difficult to approximate & example of some environment whose dynamics can be approximated well --> then the author claims that approximating environment dynamics is more difficult and some rollout-based planning should be preferred in MBRL --> propose BOOTS --> BOOTS performs well on some challenging tasks
> >
> > I do not mean "dynamics should always be considered as complicated." Of course, there could be simple environment dynamics. But how can we know that the underlying true function of environment dynamics cannot have the similar structure which is difficult to approximate as your value function example does?
> >
> > MB planning v.s. MB policy optimization. Again, this is a minor issue as I mentioned and it does not affect my assessment.
> > I personally like the concept of planning introduced in Chapter 8, page 132, from Sutton's RL book (http://incompleteideas.net/book/bookdraft2017nov5.pdf): "The word planning is used in several different ways in different fields. We use the term to refer to any computational process that takes a model as input and produces or improves a policy for interacting with the modeled environment."
> >
> > It is ok that the author has his/her own understanding.

---

> > > ### Author Response · Authors · 2019-11-15
> > > **Thanks for the reply**
> > >
> > > We appreciate the reviewer's reply and clarification very much.
> > >
> > > We would like to clarify our intended reasoning line below.
> > >
> > > 1. We started with wondering whether the distinction of model-based and model-free RL can be studied through the lens of expressivity.
> > >
> > > 2. Then we found out that indeed some constructed environments can have the property that the Q-function/policy is more complex than the dynamics.
> > >
> > > 3. But so far, it's unclear whether the phenomenon demonstrated on the toy construction is practically relevant. Do real-world environments have the property that "Q-functions are more complex than dynamics"? Certainly, perhaps many realistic environments do not have this property.
> > >
> > > 4. It's challenging to verify this property on real environments empirically, but we can try to test the usefulness of its implication --- suppose an environment has this property, then we know that we should do some rollout-based planning, such as the simple planning method BOOTS that we proposed. Assuming that the Q-function/policy is more complex than the dynamics, then BOOTS should be useful. Thus, the benefit of BOOTS is mostly to demonstrate that the theoretical understanding is practically relevant. But indeed, the benefit of BOOTS cannot fully prove real environments have more complex Q-function than dynamics. It only demonstrates the plausibility of the theory for some real environments. (In the revision, we also added an experiment in Figure 5 that shows that BOOTS+MBPO can have better asymptotic performance than MBPO.)
> > >
> > > Thus, we didn't mean that "approximating environment dynamics is more difficult [or easier] and some rollout-based planning should be preferred in MBRL." We meant that *only when* the Q-function/policy is harder to approximate than the dynamics, then rollout-based planning may be preferred. It's hard to tell whether Q-function/policy is harder to approximate, but we may optimistically try the planning algorithm.
> > >
> > > We hope our reasoning line makes sense. We will revise and clarify in a few places of the paper to make this more clear. If the reviewer thinks that some sentences in the intro or the paper may deviate from the sentiment above, it would be appreciated if the reviewer can let us know.
> > >
> > > We also appreciate the reviewer to remind us of the broader use of the term "planning" in different fields. We will clarify our use of the terms in the paper more clearly. We also modified our original response slightly to acknowledge this.
> > >
> > > Thanks!
> > > Authors

---

### Official Review · AnonReviewer1 · 2019-10-25
**Official Blind Review #2**

**Rating:** 3

**Review:**

Summary: This paper studies a theoretical aspect of the expressivity of policy, Q functions and dynamics. Based on theoretical and empirical analysis, the authors propose a new model-based RL algorithm that said to improve task performance. Final evaluations are demonstrated on MuJoCo benchmark tasks.

Overall, the paper pursues an interesting and ambitious problem on the interplay between model-based and model-free approaches, and the expressivity of the representation of dynamics, policy and value functions. However, the results in the theory part is drawn based on analysis on a very simple and special task. Therefore the theoretical results can not be considered general for all MDP cases. In addition, these results are not surprising.

- Theorem 4.3 states a general theoretical result that holds for neural networks, the proof does not look like it can hold with a universal representation power of a neural network.

- The idea of using Q-functions estimate as Boostrapping is just an idea of using on-planning to improve action selection at every decision step. This is just a recurring idea of many model-based RL approaches. BOOTS consumes more computations as planning, hence would perform better than the baselines. BOOTS should be compared to other model-based approaches that also use planning at Testing, assume all are given the same budget of testing time.

**Experience Assessment:**

I have published one or two papers in this area.

**Review Assessment: Checking Correctness Of Derivations And Theory:**

I assessed the sensibility of the derivations and theory.

**Review Assessment: Checking Correctness Of Experiments:**

I assessed the sensibility of the experiments.

**Review Assessment: Thoroughness In Paper Reading:**

I read the paper at least twice and used my best judgement in assessing the paper.

---

> ### Author Response · Authors · 2019-11-14
> **response to R1**
>
> We thank the anonymous reviewer for the review!
>
> First of all, we are glad that the reviewer appreciates the importance of studying expressivity in RL and acknowledges that this is an ambitious direction. We agreed that our results only show that there exists MDPs with the property that the dynamics is simple but the Q-function is complex. We consider the existential result as significant and meaningful because a) there is no tabular MDP with this property --- this is a unique phenomenon of the continuous state space; b) we cannot hope for a universal result because there are also MDPs without this property. Our main goal is to have a first-cut theoretical result that demonstrates that this phenomenon exists for continuous state-space problems.
>
> We address the other comments below.
>
> --- “Theorem 4.3 states a general theoretical result that holds for neural networks, the proof does not look like it can hold with a universal representation power of a neural network. “
>
> We are not entirely sure what the reviewer means here. Our theorem states that any neural network with polynomial-size cannot approximate the Q function with a near-optimal reward (that is, a constant factor within the optimal reward.).
>
> If the reviewer suspects our proof is not correct, we respectfully ask the reviewer to be more explicit about it. We note that our theorem does NOT contradict the universal approximation of neural networks, because the universal approximation may require a neural network with *exponential* width. In fact, this is the key point of the paper: any polynomial-size neural networks *cannot* approximate the Q-function in our case.
>
> --- “BOOTS should be compared to other model-based approaches that also use planning at Testing, assume all are given the same budget of testing time.”
>
> The recent benchmarking paper “Benchmarking Model-Based Reinforcement Learning” [1] shows with systematic experimentations that many model-based planning algorithms (including PETS [2], GPS [3]) don’t perform well on difficult MuJoCo tasks, especially on the most difficult Humanoid environment with long horizons. For example, as shown in Figure 5(p) in [1], none of these algorithms reaches non-trivial reward after training for 200k steps on Humanoid. (Precisely, they have rewards below 250, whereas our algorithm achieves more than 4K reward. The optimal reward is about 5K.) Therefore, we only compare with SAC and STEVE (the two algorithms that are reported to work very well for the Humanoid environment) and MBPO, which is considered to be the state-of-the-art for MuJoCo environments in sample efficiency. (Actually, our algorithm is faster than MBPO because we use a single dynamics instead of an ensemble).
>
>
> [1] Benchmarking Model-Based Reinforcement Learning. Wang et al., https://arxiv.org/pdf/1907.02057.pdf
> [2] Deep Reinforcement Learning in a Handful of Trials using Probabilistic Dynamics Models. Chua et al., https://arxiv.org/abs/1805.12114
> [3] Guided Policy Search, Levine et al., http://proceedings.mlr.press/v28/levine13.html

---

### Decision · Program_Chairs · 2019-12-19

**Decision:**

Reject

**Comment:**

The paper provides some insight why model-based RL might be more efficient than model-free methods. It provides an example that even though the dynamics is simple, the value function is quite complicated (it is in a fractal). Even though the particular example might be novel and the construction interesting, this relation between dynamics and value function is not surprising, and perhaps part of the folklore. The paper also suggests a model-based RL methods and provides some empirical results.

The reviewers find the paper interesting, but they expressed several concerns about the relevance of the particular example, the relation of the theory to empirical results, etc. The authors provided a rebuttal, but the reviewers were not convinced. Given that we have two Weak Rejects and the reviewer who is Weak Accept is not completely convinced, unfortunately I can only recommend rejection of this paper at this stage.